 FEATURE ARTICLE

EDUCATION AND OUTREACH

# March Mammal Madness and the power of narrative in science outreach

**Abstract** March Mammal Madness is a science outreach project that, over the course of several weeks in March, reaches hundreds of thousands of people in the United States every year. We combine four approaches to science outreach – gamification, social media platforms, community event(s), and creative products – to run a simulated tournament in which 64 animals compete to become the tournament champion. While the encounters between the animals are hypothetical, the outcomes rely on empirical evidence from the scientific literature. Players select their favored combatants beforehand, and during the tournament scientists translate the academic literature into gripping "play-by-play" narration on social media. To date ~1100 scholarly works, covering almost 400 taxa, have been transformed into science stories. March Mammal Madness is most typically used by high-school educators teaching life sciences, and we estimate that our materials reached ~1% of high-school students in the United States in 2019. Here we document the intentional design, public engagement, and magnitude of reach of the project. We further explain how human psychological and cognitive adaptations for shared experiences, social learning, narrative, and imagery contribute to the widespread use of March Mammal Madness.

KATIE HINDE*, CARLOS EDUARDO G AMORIM, ALYSON F BROKAW, NICOLE BURT, MARY C CASILLAS, ALBERT CHEN, TARA CHESTNUT, PATRICE K CONNORS, MAUNA DASARI, CONNOR FOX DITELBERG, JEANNE DIETRICK, JOSH DREW, LARA DURGAVICH, BRIAN EASTERLING, CHARON HENNING, ANNE HILBORN, ELINOR K KARLSSON, MARC KISSEL, JENNIFER KOBYLECKY, JASON KRELL, DANIELLE N LEE, KATE M LESCIOTTO, KRISTI L LEWTON, JESSICA E LIGHT, JESSICA MARTIN, ASIA MURPHY, WILLIAM NICKLEY, ALEJANDRA NÚÑEZ-DE LA MORA, OLIVIA PELLICER, VALERIA PELLICER, ANALI MAUGHAN PERRY, STEPHANIE G SCHUTTLER, ANNE C STONE, BRIAN TANIS, JESSE WEBER, MELISSA WILSON, EMMA WILLCOCKS AND CHRISTOPHER N ANDERSON

*For correspondence: katiehinde@gmail.com

## Introduction

Public education and outreach are an essential pillar of 21st century scholarship. A substantial portion of empirical research and research infrastructure, especially in higher education, is supported through public funds. Research output is therefore not only expected to serve the public good (*Hazelkorn and Gibson, 2019*), but a broad view of the social contract conceptually situates scientific knowledge generated with public funds within the public trust (*Schroeder et al., 1989*; *Gibbons, 1999*; *Hetland, 2017*; *Krishna, 2020*; for important exceptions, see *Fox, 2020*). Advocacy for Open Science has grown in recent decades (*Sá and Grieco, 2016*; *Cribb and Sari, 2010*; *Piwowar et al., 2018*; *NASEM, 2018*) but even when scholarly publications are open access, empirical findings too often remain behind a paywall of jargon. As such, institutions, funding agencies, professional societies, and individual scholars increasingly recognize the importance

of science communication (hereafter SciComm) and informal STEM education to reach learners, clinicians, policy-makers, and other members of the general public (*Beaulieu et al., 2018*; *Jessani et al., 2018*; *Bell, 2016*; *National Science Board, 2011*; *Yuan et al., 2019*). Moreover, increased visibility of science and scientists can counter stereotypes about who does science and inspire the next generation of scientists (*Woods-Townsend et al., 2016*; *Jarreau et al., 2019*).

Across the life, biomedical, physical, and social sciences, scholars participate in SciComm and educational outreach (*Yuan et al., 2019*; *Cooke et al., 2017*), and increasingly leverage social media platforms to achieve these broader impacts (*Bik et al., 2015*; *Collins et al., 2016*; *McClain and Neeley, 2014*; *Mehlen-bacher, 2019*; *Jarreau et al., 2019*). SciComm and educational outreach campaigns, however, can be variably successful in their content, reach, propagation, and sustainability and "impact" is often opaquely operationalized, measured, or assessed (*Saunders et al., 2017*; *Davies, 2019*). Web traffic, social media engagement, and long-term use of resources are most often used as indicators of SciComm impact (*Saunders et al., 2017*; *Fernández-Bellon and Kane, 2020*). Comprehensive roadmaps of successful SciComm initiatives, campaigns, and programs have been infrequently described in the scholarly literature. Early and recent reports, however, have demonstrated that memes, images, activities, and dynamic content from scientists are associated with increased learner and public interest, competencies, donations, and enthusiasm for nature (*Moskal et al., 2007*; *Hone et al., 2011*; *McClure et al., 2020*; *McClain, 2019*; *Lenda et al., 2020*).

Our SciComm program March Mammal Madness (MMM) engages hundreds of thousands of members of the general public in a celebration of animal behavior, and the broader natural world, for several weeks each year. Notably, March Mammal Madness blends together four approaches to science outreach – gamification, social media platforms, community event(s), and creative products (*Subhash and Cudney, 2018*; *Varner, 2014*; *Bush et al., 2018*) – with salient animal-based content. Science communicators have previously recognized that students in the United States are particularly interested in animal behavior (*Bush et al., 2018*) across urban, suburban, and rural landscapes in which species diversity and visibility varies (*Schuttler et al., 2019*). At very young ages, children are attracted to neotenous and familiar animal phenotypes (*Borgi et al., 2014*; *Borgi and Cirulli, 2015*). Children and young adults also express greater affinity for mammals and birds than reptiles, insects, and amphibians (*Schlegel and Rupf, 2010*). Leveraging the dynamic game elements of a single elimination tournament combined with story-telling scientists, March Mammal Madness makes accessible reports from the scientific literature including elegant behavioral ecology experiments (*Morand-Ferron et al., 2016*; *Campbell et al., 2009*), meticulous natural history descriptions (*Able, 2016*; *Tewksbury et al., 2014*), and gripping narratively-constructed accounts of observed animal behavior (*Ramsay and Teichroeb, 2019*).

The tournament also provides lesson plans as an Open Educational Resource (*Miao et al., 2016*) to educators who systematically integrate March Mammal Madness into their curriculum. March Mammal Madness achieves key SciComm goals by reaching many audiences (*Varner, 2014*), facilitating interactions between scientists and students (*Boyette and Ramsey, 2019*), and effecting propagation and sustained adoption of the tournament (*Stanford et al., 2017*). Across 11 evenings, beginning with a Wild Card through early rounds into the Sweet Sixteen, the Elite Trait, the Final Roar, and finally the Championship "battle", March Mammal Madness is a SciComm extravaganza.

Here we systematically document our intentional design, magnitude of reach, and compounding impact of March Mammal Madness. We further contextualize how human psychological and cognitive adaptations for games, shared experiences, co-constructed narratives, and artistic illustration likely underlie the sustained success of this science communication approach. We posit that March Mammal Madness models generalizable and scalable tactics for other scientists seeking to develop or expand their own science communication. Alternatively, and with much less effort, scientists can incorporate March Mammal Madness into their own outreach portfolio by introducing the tournament into their labs, classrooms, and communities.

*"This was no ordinary death,*
*though forty million years*
*lay between us and that most gaping snarl.*
*Deep-driven to the root a fractured scapula*
*hung on the mighty saber undetached; two beasts*
*had died in mortal combat, for the bone*
*had never been released"*

Excerpt from poem "The Innocent Assassins" (*Eiseley, 1973*). Loren Eiseley wrote this poem about an inferred battle between two Nimravids that ended in mutual destruction, a fossil discovery that was first described by *Toohey, 1959*.

## March Mammal Madness

Each March, dozens of academics, conservationists, and artists use the social media stage of Twitter to deliver performance science in the form of a simulated tournament to reveal an annual animal champion (*Figure 1*). Each year, we release a unique bracket revealing the selected combatants organized into four thematic divisions. Players predict the likely outcomes of sequential encounters between pairs of combatants based on the player's knowledge, preferences, or taxon allegiances. After allowing players ~ 10 days of research to make bracket predictions, the official tournament outcomes are revealed over several weeks using science-based story-telling. Scientist-narrators "live-announce" the crafted encounters like a sporting event radiocast on the social media platform Twitter, as players follow along, primarily via mobile devices (53%) or desktop/laptop computers (41%). Scientist-narrators typically use a standardized narrative arc, in sequence presenting background "stats" for each combatant, describing the scene of the "battle," and then creatively report the back-and-forth details of the encounter like a sports play-by-play (see *Supplementary files 1* and *2*).

Although rife with pop culture jokes and internet memes, March Mammal Madness is systematically anchored to the scientific literature (*Hinde et al., 2017*; *Fisher, 2018*). For each simulated battle, scientist-narrators provide key information about each combatant species and feature facts about behavior, life history, conservation status, phylogeny, morphology, and other exceptional adaptations. Predation tactics, anti-predator defenses, kleptoparasitism, kill ownership, maternal aggression, signaling behavior, optimal foraging, interspecific displacement, sickness behavior, winner effects, gut passage time, and many other aspects of animal behavior, physiology, and morphology are routinely invoked in battle narrations, often with specific citations linked. Additional facts and images are tweeted by geneticists and partner organizations such as the American Society of Mammalogists, Cleveland Museum of Natural History, and the Aldo Leopold Foundation. Immediately after the evening's battles conclude, written "sports summaries" of the battles (see *Supplementary file 3*) and underlying science and full transcripts of the play-by-play are posted on multiple online platforms including Facebook, Wakelet, Blogspot, and LibGuide so the science behind the outcomes is widely available. These materials are additionally distributed directly to educators using March Mammal Madness in their classrooms so student players can follow the tournament without being on social media or accessing the internet. Our tournament motto perennially emphasizes "If you're learning, you're winning."

### Tournament species

March Mammal Madness has featured hundreds of species from a global distribution of biogeographic regions (N = 383 species across 2013–2019). Combatants have represented N = 25/27 mammalian orders, all except for Paucituberculata and Microbiotheria. Species inclusion as tournament combatants, however, does not achieve proportional representation across mammalian orders (*Burgin et al., 2018*), much to the oft-communicated ire of researchers studying Chiroptera. Carnivora, Artiodactyla, and Diprotodontia are particularly over-represented as tournament combatants (*Figure 2*) and taxa from these orders have more often been featured in two or more tournament years as repeat entrants. Chiroptera, Rodentia, and Eulipotyphla are consistently featured as combatants, but have been under-represented in proportion to their actual species counts, while small-bodied taxa from mammalian orders less familiar to the general public have been routinely showcased (*Figure 2*). As such, each year our bracket includes well-recognized charismatic megafauna, familiar backyard species, and introduces rare taxa many players have never encountered in their zoo visits, reading, or nature program viewing.

Although the tournament particularly celebrates Class Mammalia, many non-mammal combatants have been included in March Mammal Madness; N = 53 in total from 2013 to 2019. While early tournaments only showcased a smattering of non-mammals, since 2018 March Mammal Madness has featured dozens of diverse

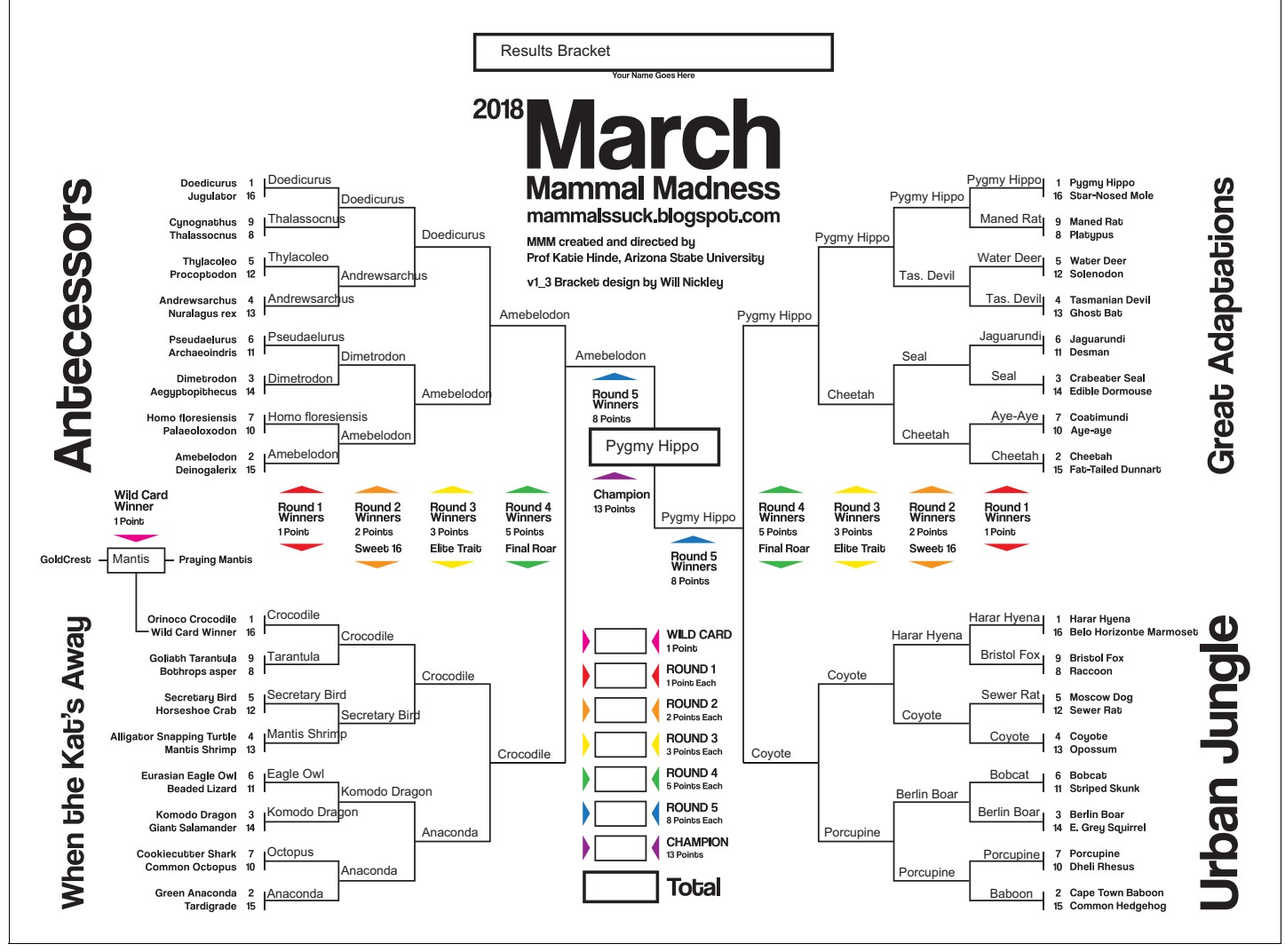

**Figure 1.** The tournament outcome bracket for March Mammal Madness in 2018. Players initially begin with a "blank" bracket listing just the first-round match-ups and predict sequential match outcomes from their pre-existing knowledge, targeted research, and/or guessing. In the 2018 tournament the four divisions were the 'Antecessors' (fossil species that "came before" today's living mammals, stretching back to the synapsids), 'Great Adaptations' (mammals that have exceptional and rare traits), and 'Urban Jungle' (mammals that survive, and sometimes thrive, in suburbs and cities). The last division, 'When the Kat's Away', was a colloquial allusion to entomologist Chris Anderson and ichthyologist Josh Drew inserting a division of non-mammal combatants for the launch of the tournament when mammalogist Katie Hinde was out of the country. In the Final Four, elephant-relative *Amebelodon* emerged victorious from the Antecessors and defeated #AltMammal Orinoco crocodile, but was wounded during the encounter. Coyote may have been king of the Urban Jungle but was no match for the pygmy hippopotamus (from Great Adaptations). In the ultimate showdown, *Amebelodon*'s larger size and weaponry could not overcome his previously-sustained injuries, and he was displaced by surprise 2018 Champion pygmy hippopotamus.

animal taxa including insect, amphibian, lepidosaurian, archosaurian (including avian), cephalopod, arachnid, crustacean, and tardigrade combatants. In an effort to further expand the topics included in our science outreach and to intentionally disrupt "plant blindness" (*Jose et al., 2019*), we included several plant species in 2019. Organismal diversity and description have waned as foci within biology curricula, in tandem with decreases in student and public engagement with nature (*Tewksbury et al., 2014*; *Greene, 2005*; *Schmidly, 2005*). By structuring the tournament around organisms and routinely linking to the higher and lower levels of biological complexity (*Greene, 2005*), March Mammal Madness continuously spins a sparkling kaleidoscope of biological life on earth.

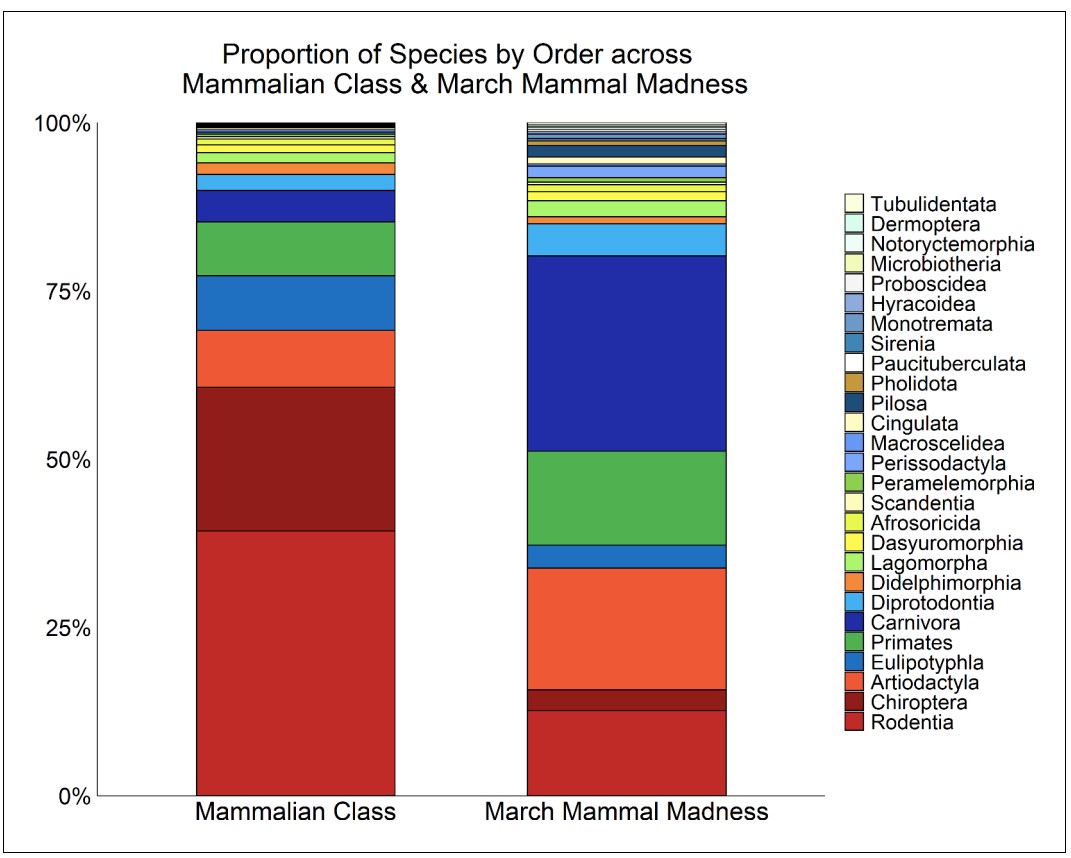

**Figure 2.** How the combatants featured in March Mammal Madness compare with mammals in general. Proportion of extant species by order across the mammalian class, stacked according to the species count of the order (with the largest order at the bottom; left), and as combatants in March Mammal Madness (right). Some orders (such as Rodentia) have been under-represented in MMM (reds), some are over-represented (such as Carnivora; blues), and others have been proportionately represented (yellows).

### Tournament divisions

Each year, March Mammal Madness presents combatant species in four novel "Divisions" (*Table 1*). In the inaugural year, the divisions were largely organized around mammalian Orders (Carnivora, Primates), that had the dual drawbacks of reduced phylogenetic representation across the mammalian Class and substantial redundancy of attributes among many combatants due to recent shared common ancestry. Since 2014, we have intentionally designed divisions to integrate more complex themes of environments, extinction-risk, adaptations, lexical quirks, among other bins. These divisions demonstrate how biological life can be clustered according to diverse taxonomies (*Medin and Bang, 2014*) and facilitate dialogues about historical context of scientific "discovery." For example, in 2019 the CAT-e-GORY Division featured many "cool cats," but no species from the mammalian Family Felidae. Rather these were taxa whose English common name or scientific

binomial alluded to phenotypic similarities to felids, an extensively used comparand in common names and taxonomic nomenclature. This division provided important opportunities to highlight the intertwining of scientific colonialism, linguistic privilege, and phylogenetics as the co-occurrence of European Imperialism and the formalization of Linnean taxonomy manifested in a rapid global cataloging of fauna (*Raj, 2000*; *Smith and Jackson, 2006*).

A mythical mammal division in 2015 stirred controversy as some fans initially averred the inclusion of imaginary species subverted scholarly credibility and competitively inhibited legitimate animals. Discussion of mythical mammals, however, was harmonious with the tournament's science communication priorities. Importantly, mythical mammals often feature traits or combinations of traits of species within a local ecology that present danger, risk, or usefulness to humans (*Scalise Sugiyama, 2001*), allowing narrators to include information on multiple actual

**Table 1.** Each annual March Mammal Madness tournament featured novel divisions that showcased diverse taxa.

| Year | Divisions | Description | Example taxa |
|---|---|---|---|
| 2013 | Carnivores | Meat-eaters | Lion, Wolverine |
| | Primates | Primate Order | Orangutan, Uakari |
| | Browsers and Grazers | Herbivores | Tapir, Moose |
| | Hodge Podge | Miscellaneous taxa | Wombat, Flying Fox |
| 2014 | Marine Mammals | Adapted to marine ecosystems | Narwhal, Harbor Seal |
| | Social Mammals | Highly social species (battle as a team) | Hyena, African Wild Dogs |
| | The Who in the What Now | Lesser-known taxa | Dhole, Saiga |
| | Fossil Mammals | Extinct taxa from the fossil record | Mastodon, Dire Wolf |
| 2015 | Mighty Minis | Smol bois | Bumblebee Bat, Tenrec |
| | Critically Endangered | IUCN red list taxa | Iberian lynx, Tenkile |
| | Sexy Beasts | Traits strongly influenced by sexual selection | Irish Elk, Elephant Seal |
| | Mythical Mammals | Creatures from cultural myths and folklore | Minotaur, Yeti |
| 2016 | Cold-adapted | Adapted to cold environments/seasons | Snow Leopard, Caribou |
| | Mighty Giants | Large in size/for their clade | Panda, Giant Armadillo |
| | Mascot Mammals | Mascots of colleges/universities | (Howard) Bison |
| | Mammals of the Nouns | Ecosystem niche featured in common name | 'Hyrax of the Rock' |
| 2017 | Desert-adapted | Adapted to arid environments | Aardwolf, Saiga |
| | Coulda Shoulda | Contenders defeated unexpectedly 2013–16 | Sabertooth Cat, Lion |
| | Adjective Mammals | Common name includes adjective | Sac-winged Bat |
| | Two Animals, One Mammal | Taxa with two-part animal common names | Spider Monkey |
| 2018 | Antecessor | Synapsids and their fossil descendants | *Dimetrodon*, *Doedicurus* |
| | Great Adaptations | Unique/exceptional traits | Crabeater Seal, Aye Aye |
| | Alt-Mammals | OK FINE, WE'LL HAVE NON-MAMMALS | Mantis Shrimp, Secretary Bird |
| | Urban Jungle | Taxa that thrive in high density human areas | Coyote, Rhesus |
| 2019 | Waterfalls | Aquatic adaptations | Aquatic Genet, Manatee |
| | Tag Team | Inter-species mutualisms (battle as a team) | Banded Mongoose and Warthog |
| | Jump-Jump | Adaptations for saltation | Jackrabbit, Serval |
| | CAT-e-GORY | Nomenclature referring to a felid | Sea Lion, Tiger Owl |

species in tandem with the mythical stories. For example, one contestant was the ichneumon, a mythical mammal which would allow itself to be swallowed by a crocodile and then burst out, and in doing so, would kill its sworn enemy (*Budge, 1969*). Through this myth, we were able to not only showcase the role of crocodiles as apex predators in African river systems, but also introduce the biology of ichneumon wasps – a group of insects that lay eggs within other insect species, the larvae hatch within and emerge, thereby killing the host (*Gauld and Bolton, 1988*). Tales of magic beings or objects have the greatest diffusion across cultural landscapes and can persist for thousands of years (*da Silva and Tehrani, 2016*). As such, mythical creatures can serve as valuable symbols around which conservation themes can be structured (*Holmes et al.,*

*2018*). Myths and mythical mammals as phenomena are constructs that emerge from human adaptations for social learning, credulity, and abstract thought, allowing MMM to reflexively discuss how evolution has shaped humans (*Ihejirika and Edodi, 2017*, *Barrett et al., 2016*; *Kline, 2015*). Lastly, by including mythology gleaned from antiquity and ethnography, we hoped to broaden participation among students and scholars in the humanities.

Within divisions, combatants are assigned relative rankings, termed "seeding", that suggest expected competitiveness within the tournament construct (*Schwenk, 2000*). Seedings are largely based on upper limits of combatant mass, with predators "punching above their weight." Seed assignment can be, in part, to facilitate more reasonable first round match-ups in terms of

battle substrate (terrestrial vs. aquatic match-ups are typically avoided in the first round) or to minimize counter-productive digressions in classrooms of adolescents in contexts of various cultural sensibilities (*Skiba et al., 2016*). For example, one year our initial seed assignment would have generated a macaque vs. deer match-up shortly after extensive media coverage of inter-specific sexual behaviors between *Macaca fuscata* and *Cervus nippon* (*Gunst et al., 2018*), prompting seed re-assignment early in tournament planning. Once we finalize the full bracket line-up, the MMM scientific team conducts additional research to evaluate likely match outcomes, accounting for battle ecology. Following team evaluation and discussion, outcome probabilities are assigned to each match-up. These probability estimations are used in conjunction with a 1-100 random number generator to determine the "official" match outcomes and allows the random occurrence of upsets (see Battle Outcomes below). The scientist-narrators then use the scientific literature or personal experiences in the field to craft plausible battle scenarios. In this way, the tournament incorporates structured game mechanics around science learning (*Subhash and Cudney, 2018*).

### Battle location (This is not Thunderdome)

Battle narrations are situated across diverse ecosystems globally and are March Mammal Madness canon. Early rounds of the tournament favor the better-ranked combatant by situating the encounter in their own habitat, a "home-court advantage" that potentially disadvantages their opponent. Adaptations mismatched with ecological context have contributed to tournament losses due to hyperthermia (*Panthera uncia, Gulo gulo*), hypoxia (*Mustela erminea*), and osmotic imbalance (*Octopus vulgaris*). More advanced rounds – the Elite Trait, the Final Roar, and the Championship – are randomized among four possible ecosystems specific to each tournament year (*Figure 3*). Scientist-narrators often situate battles in specific locations to highlight national parks, conservation areas, public lands and/or endangered ecosystems (*Bland et al., 2017*). Tournament spectators have been figuratively transported to the Karakum Desert in Turkmenistan; Gunung Leuser National Park, Indonesia; the Cojedes River, Venezuela; Bears Ears National Monument, USA; coastal ice flows of Antarctica; Cradle Mountains-Lake St. Clair National Park, Australia; Namib-Naukluft National Park, Namibia; and thorn forests of the Deccan Plateau, India, among hundreds of other locations. Figurative transportation has been combined at times with time travel, as battles involving fossil combatants occur within specific paleoenvironments. For example, a battle between *Andrewsarchus mongoliensis* and *Nuralagus rex* took place 40 million years ago in a humid forest in what is present-day Inner Mongolia. Scientist-narrators frequently highlight aspects of the community ecology, particularly carnivore guilds that have shaped the evolution of the combatant species (*Caro and Stoner, 2003*). Of additional interest in the tournament are ecosystem engineers whose activities alter physical structures within the environment,

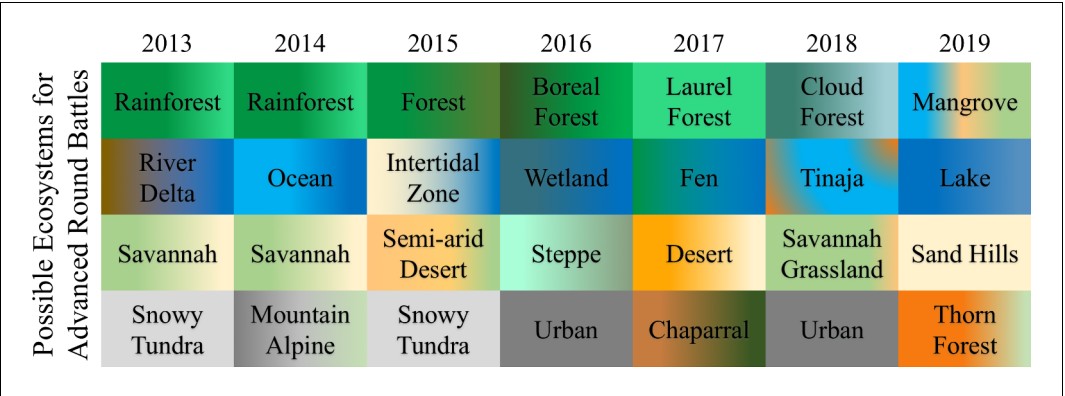

**Figure 3.** Battles in the advanced rounds of the tournament take place in one of four randomly selected ecosystems. The four ecosystems or habitats that might be used in the advanced rounds of the tournament (that is, in the four Elite Trait battles, the two Final Roar battles and the Championship battle) are announced during the pre-season, with the ecosystem to be used being revealed in "real time" during the play-by-play narration. Colors are largely indexical to represent predominant hue(s) within the ecosystem. Generally, greens represent forest, blues represent aquatic systems, ochres represent scrublands and sandy deserts, and gray represent urban spaces.

impacting numerous other taxa (*Coggan et al., 2018*).

> *"Beaver ponds are prime habitat for Mink's preferred meal... MUSKRAT (*Crego et al., 2016*). Beaver brings all the Mink to the yard, because their Muskrat, it's better than yours. Dam right, it's better than yours #BeaverDamPond #2019MMM"*
> —Scientist-Narrator Tweet

Impacts of the human-driven global climate crisis, such as extreme sea ice retreat (*Durner et al., 2011*), permafrost thaw-slumping (*Wang et al., 2014*), and range constriction on altitudinal gradients (*Henry et al., 2012*) have been decisive factors in battle outcomes. Narrations have further stressed that in addition to the humanitarian devastations associated with human conflict, warfare has significant, though poorly understood, ecological impacts (*Machlis and Hanson, 2008*).

## Battle outcomes

The conclusion of these imaginary encounters among tournament combatants typically fall into three general domains; "Red, in tooth and claw" (to quote from "In Memoriam A.H.H." by Lord Tennyson), "the better part of Valour, is Discretion" (from *Henry IV, Part 1* by Shakespeare), and Deus ex Machina (*Figure 4*). Lethal or devastating injuries can occur from predation, anti-predator defense, territorial encounters, or conflict over a recent kill, and were coded as a "technical knock out" (TKO). Scientist-narrators have described apex predators' mortal attacks on mesopredators, parental defense of young, and other intentional conflicts that escalated into physical attacks. TKO outcomes occurred in ~50% of tournament battles (N=225/451). But in nature the injury risks and/or energy costs associated with physical attacks, when weighed against potential benefit, can frequently precipitate de-escalation, retreat, or withdrawal (*Parker and Rubenstein, 1981*; *Archer et al., 1994*; *Briffa and Sneddon, 2007*), outcomes often intentionally featured in March Mammal Madness (32%, N=146/451).

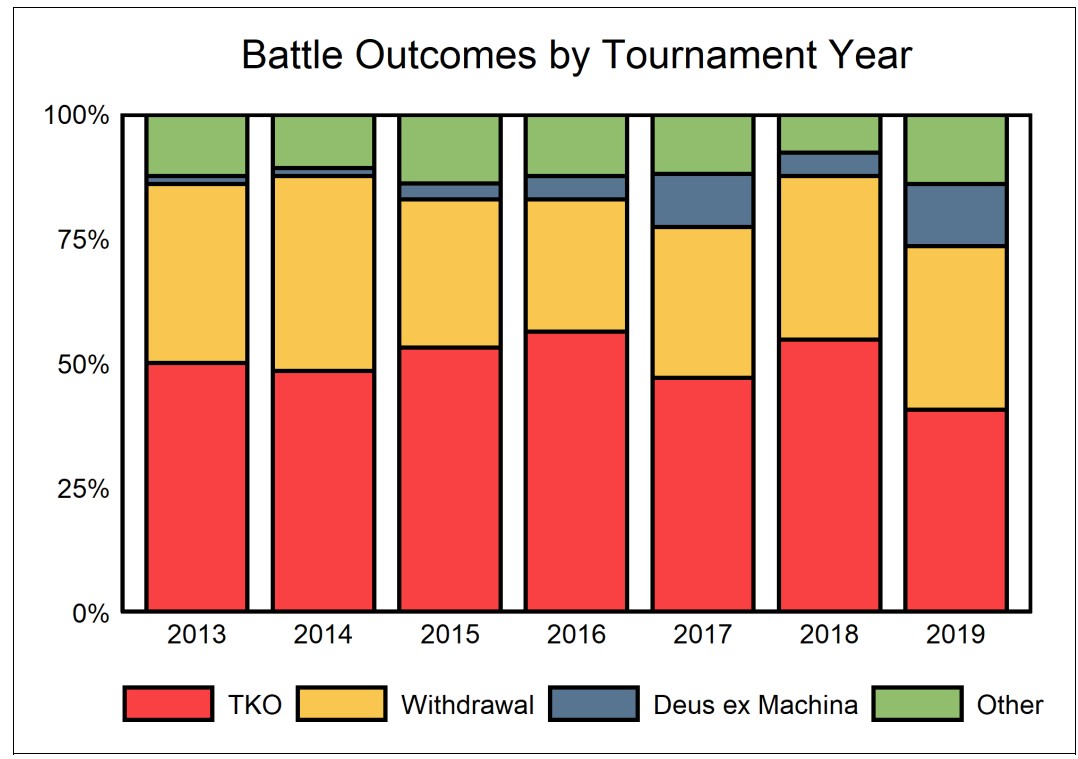

**Figure 4.** How battles end in March Mammal Madness. Most battles conclude with a fatal or debilitating encounter between the two combatants (also known as a technical knock out or TKO). Withdrawals from the encounter are also common, as are third-party interventions (Deus ex Machina) that cause one combatant to advance in the tournament.

The device of Deus ex Machina, resolution via an unexpected and external agent, is used by scientist-narrators to highlight important sources of mortality for species, account for improbable outcomes forced by improbable outcome randomization, or to diversify story arcs across battles. While only a small proportion of outcomes (5.5%, N=25/451), the Deus ex Machina device often incenses players, but suggests particularly strong long-term retention of information. For example, in 2014 in a 1st-round battle between a fossa (*Cryptoprocta ferox*) and a pangolin (*Manis crassicaudata*), a poacher collected the defensively curled pangolin for illegal animal trafficking. The day before this battle was live-tweeted, the IUCN Pangolin working group reported pangolins as the most trafficked animal globally (*Zhou et al., 2014*), hence making for not only a topical and timely narrative, but a 3rd party intervention that players continue to spontaneously bemoan years later.

"Other" outcomes (12%, N=55/451) featured in March Mammal Madness battles include prioritization of foraging, dam-building, nest relocation, distraction by mating competition, electrocution (*Kumar and Kumar, 2015*), Takotsubo cardiomyopathy (*Blumstein et al., 2015*), foraging exclusion, displacement, and cryptic hiding. Typically, the better-seeded species defeated the worse-seeded species, but on average 22% (mean=13 ± 2.2 sd) of battle outcomes involved an "upset" in which the worse-seeded species advanced. In the NCAA men's basketball March Madness tournament, historically ~22% of outcomes have been characterized as "upsets" (*Greenburg, 2019*). We do note, however, that the NCAA definition of upset is more conservative in terms of relative rankings – 2 or more seeds distant – as is appropriate for a more evenly-matched tournament in which all participants are of the same species.

Events occurring in one round are carried forward in a combatant's story arc. Combatants advancing in the tournament have had to grapple with snapped canines, wrenched knee joints, wound infections, envenomations, and zoonotic disease transmissions. Scientist-narrators even account for gut passage time since last meal when describing motivation for predation.

> *"Having gorged on capybara only yesterday, Coyote & Badger are "full & lazy" as happens to carnivores on "many days of their lives" (Jeschke, 2007) #2019MMM"*
> —Scientist-Narrator Tweet

Winner effects may manifest, if the aggressive encounter involves a well-matched opponent and the combatant retains home court advantage (*Fuxjager et al., 2009*; *Huang et al., 2011*). At times battle narrations have made use of cliffhanger devices. For example, after defeating a tiger salamander (*Ambystoma tigrinum*), a fisher (*Pekania pennanti*) was trapped and transported to the Calgary Zoo. In the next battle, the audience learned the combatant had become a part of the Cascades Fisher Reintroduction Project and relocated in time for their next battle in Mt. Rainier National Park (*Lewis, 2017*). In this way, story arcs are built across the weeks of the tournament as the fandom cheers and jeers underdogs, dark horses, scaredy-cats, lone wolves, and long shots, as would-be champions experience triumph or trouncing on this figurative field of battle.

Tournament champions are most typically apex predators or large-bodied herbivores – African elephant (*Loxodonta africana*, 2013), spotted hyena clan (*Crocuta crocuta*, 2014), Sumatran rhinoceros (*Dicerorhinus sumatrensis*, 2015), tundra wolf (*Canis lupus occidentalis*, 2016), middle Pleistocene short-faced running bear (*Arctodus simus*, 2017), pygmy hippo (*Choeropsis liberiensis*, 2018), and Bengal tiger (*Panthera tigris tigris*, 2019). To date, a non-mammal has yet to achieve tournament champion, a state of affairs entirely due to empirically-grounded probabilities within the tournament structure and certainly not due to taxonomic biases (*Batt, 2009*; *Schlegel and Rupf, 2010*) that influence research effort and the scholarly literature (*Jarić et al., 2019*; *Bezanson and McNamara, 2019*) or the tournament architect.

### Battle artwork

Eleven artists have created N = 669 depictions of combatant species for the March Mammal Madness tournament. After playing the tournament in 2014, tattoo artist and scientific illustrator Charon Henning approached the narrators and offered to contribute artwork of the combatants. In 2015, Henning joined MMM leadership as tournament art director. Artists have used both digital approaches and traditional illustration media, including graphite, pen and ink, scratchboard, and acrylic paints, to depict each of the competitors (*Figure 5*). Artists created individual illustrations for each competitor for their tournament debut, and a "victory" illustration with each advance in the tournament.

First round artwork has generally been produced with a minimum of detail, while illustrations for advances became sequentially more refined. As a result, by the completion of the tournament, the champion competitor has been depicted in seven individual illustrations. Beginning in 2016, the championship portrait has been an art fusion with contributions from each illustrator involved in that year's tournament.

Using the Latin binomials, artists conduct illustration research and at times consult scientist-narrators for further information on a given species. Academic publications, species experts, and museum resources are valuable and necessary components in creating accurate and compelling illustrations. In 2015, the Critically Endangered Division presented challenges due to the dearth of photographic reference material. Many species in this division were only known from museum collections and antiquated scientific illustrations, requiring time-intensive cross-referencing with closely related species to better understand life-like appearances of these species. The art pieces for this division, however, were particularly notable for the inspired idea to incorporate extinction threat elements into the art pieces. All revenue generated by the sale of tournament artwork through the Society6 shop (https://society6.com/mammalmadness) is equitably divided among the artistic team.

### Scholarly content in battle narrations

The descriptions of species and environments and explanations of encounters that are provided in the "battles" of March Mammal Madness rely extensively on the academic literature. Since the tournament's inception in 2013 until the 2019 Championship, March Mammal Madness battles included citations to N = 1078 scholarly sources, including N = 1016 peer-reviewed journal articles from N = 350 journals. The number of scholarly publications cited each year has generally increased across the tournament years (*Figure 6A*), showing marked increases in conjunction with expansions of the narration team in 2014 (N = 4 scientist-narrators) and 2017 (N = 11 scientist-narrators). The *Journal of Mammalogy*, *PLoS One*, and the *Journal of Zoology* are most frequently cited by scientist-narrators, and many other animal-focused and general science journals are represented among the top-cited journals in March Mammal Madness (*Figure 6B*). The majority of scholarly sources, N = 689 (64%), were published in the 21st century (*Figure 6C*), but some citations included writings dating back to the 1700s

including important germinal studies of animal behavior and natural history (*Burghardt, 2020*). Naturalists' detailed, integrative descriptions of behavioral and physical characteristics are excellent for crafting MMM narratives, although experimental and explanatory science has increasingly displaced descriptive natural history, a significant loss to science and society that has been decried for decades (*Tewksbury et al., 2014*; *Greene, 2005*; *Schmidly, 2005*). Empirical citations with amazing, but real facts, can be instrumental for substantiating narrative outcomes in hotly-debated MMM match-ups that generate intense emotions among players. Primary literature can often reveal important natural history that is often elided in the online sources typically used by tournament players researching their bracket predictions. For example, many players had high hopes for the platypus upon discovering during pre-tournament research that the platypus is one of the rare venomous mammals. But during the battle play-by-play, followers were astonished to learn that platypus venom varies seasonally.

> "But platypus mating season is over and now his venomous spurs are shooting blanks! Indeed, March is when the crural glands that produce platypus venom ARE MOST SHRUNKEN AND USELESS (*Grant and Temple–Smith, 1998*) #2018MMM"
> —Scientist-Narrator Tweet

The scholarly contributions extend beyond the official narration tweets. Beginning in 2015, the American Society of Mammalogists (ASM), via the Informatics Committee, has systematically featured 241 unique photographs of combatant taxa from the ASM Mammal Images Library. As a nonprofit, educational program of the society, the Mammal Images Library is a curated collection of >4700 high-resolution images of extant and extinct mammalian species. These images, expertly identified to current taxonomy, are freely available for educational use at the ASM website, mammalsociety.org. Since 2016, Professors Anne Stone and Melissa Wilson contributed tweets featuring genetic and phylogenetic information about combatants citing an additional ~175 sources annually (*Figure 6A*). March Mammal Madness allows scientists to translate scientific academese directly in accessible, dynamic narration paired with exquisite illustration. In so doing, we reach a broader distribution of the next generation and

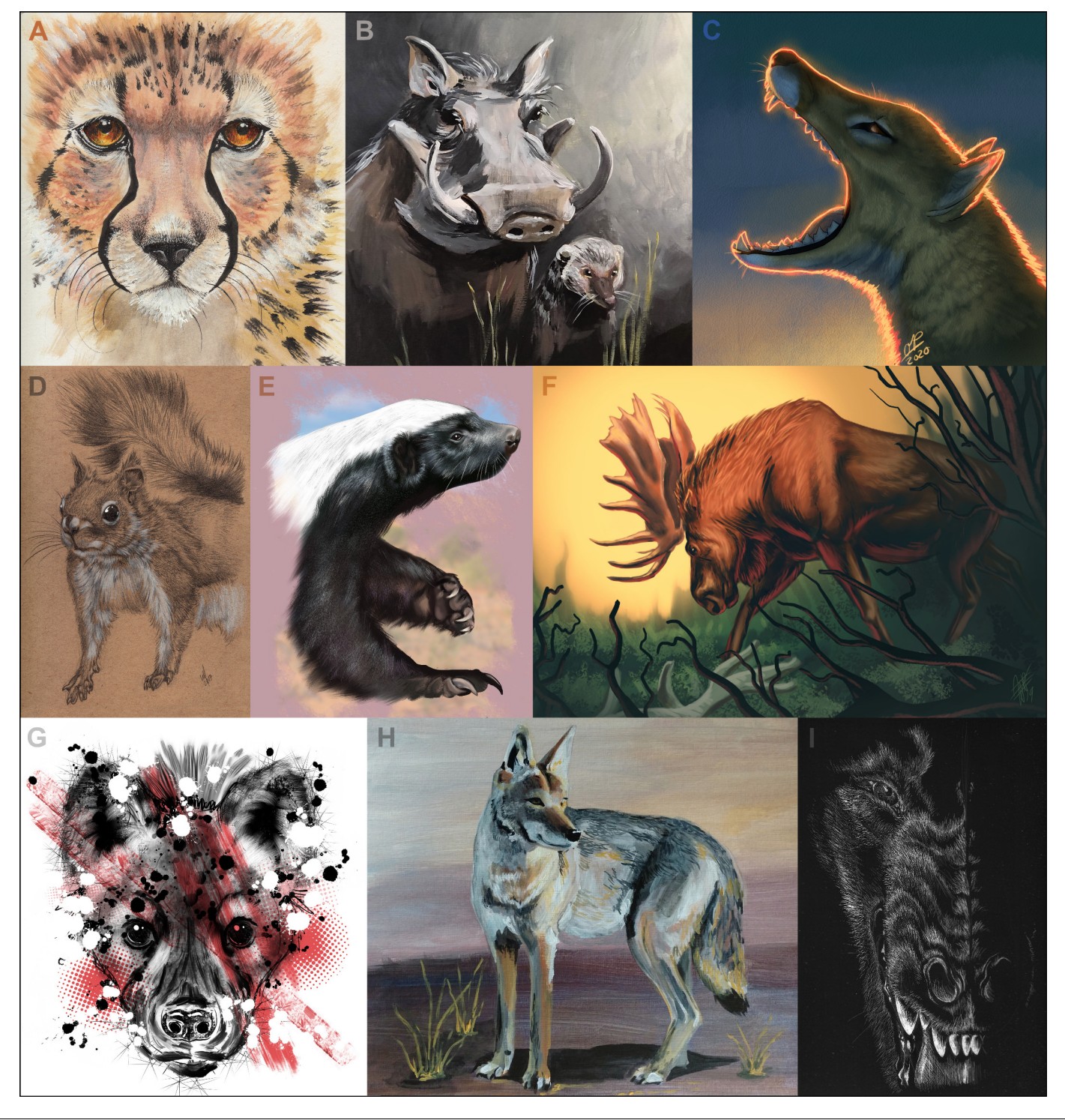

**Figure 5.** Artistic representations of some previous tournament combatants. (**A**) Cheetah by Charon Henning [http://www.charonhenning.com/]; (**B**) Tag Team Mutualists, the warthog and the mongoose, by Mary Casillas [marycasillas.wix.com/paintings]; (**C**) Thylacine by Olivia Pellicer [opellisms.com]; (**D**) Red squirrel by Charon Henning; (**E**) Honey badger by Charon Henning; (**F**) Moose by Valeria Pellicer [http://www.vpellicerart.com/]; (**G**) Spotted hyena by Charon Henning; (**H**) Coyote by Mary Cassilas; (**I**) *Andrewsarchus mongoliensis* by Charon Henning.

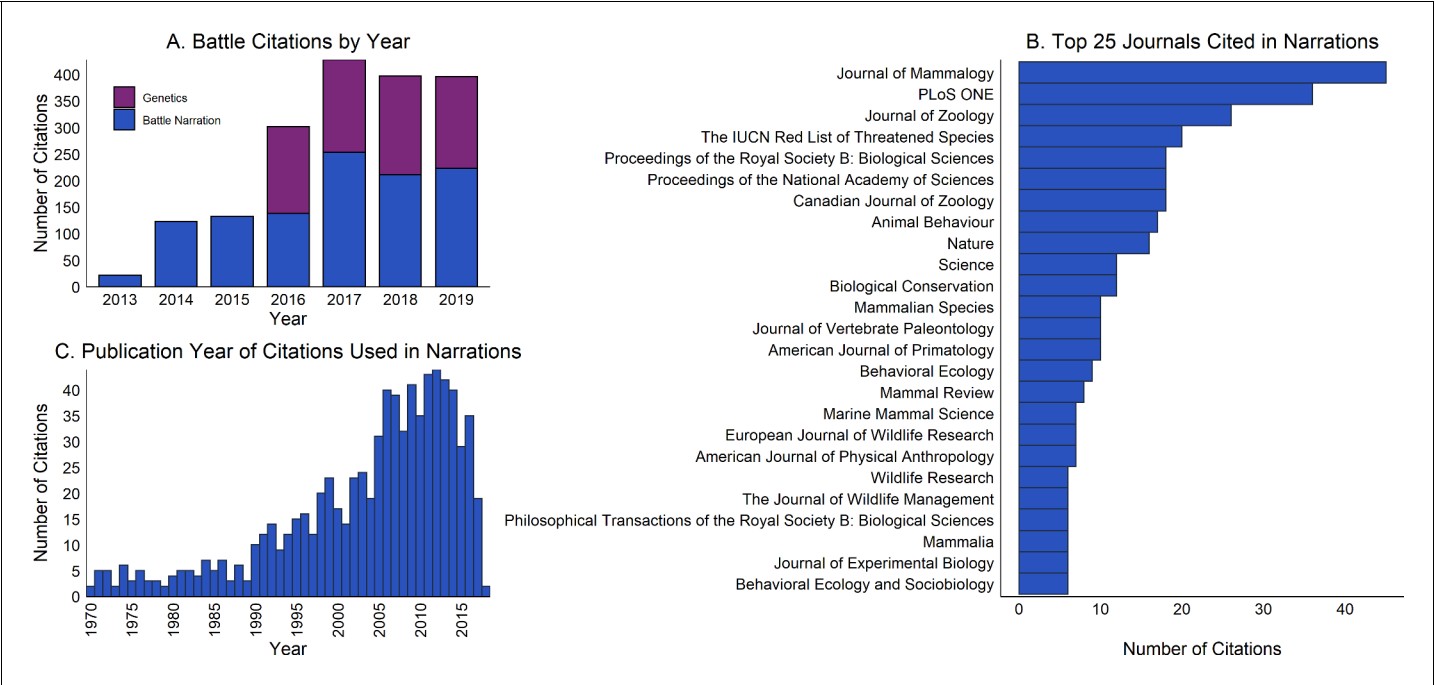

**Figure 6.** The scientific literature within March Mammal Madness. (A) During the tournament, hundreds of citations from the scholarly literature are embedded in play-by-play battle tweets from the scientist-narrators and introductory and RIP tweets from the genetics team. (B) The top 25 journals cited in the battle narrations. (C) Most of the papers cited in the battle narrations were published after 2000.

more effectively inspire appreciation for the vivid splendor of the natural world.

### Timeline, teams, and skillsets

Compelling, infectious, far-reaching SciComm is not created de novo, but rather is built cumulatively through intentional design, considered expansion, transdisciplinary collaboration, and no small amount of serendipity. Although initially created in 2013 as a reaction to a non-science based animal bracket (*Cole, 2015*) and for psychological resilience in light of other scholarly activities (*Clancy et al., 2014*; *Nelson et al., 2017*), March Mammal Madness has grown substantially from its inaugural year. In response to player and educator feedback and volunteered expertise, we have refined and expanded the tournament offerings each year (*Figure 7*). Biological anthropologists, evolutionary biologists, entomologists, mammalogists, marine biologists, paleoanthropologists, primatologists, and wildlife biologists have been instrumental, individually and in teams, in crafting battle narratives for the "performance science" of live tweeting the play-by-plays [Anderson, Brokaw, Chestnut, Connors, Dasari, Drew, Durgavich, Hilborn, Hinde, Kissel, Lee, Lewton, Light, Murphy, Tanis, Wilson, Varner] with varying amounts of input

from Editors [Anderson, Hinde]. As the narration team has grown, team members alternate serving as back-channel stage manager to direct the complex sequence of ordered battles on Twitter each tournament night.

In addition to the geneticists, professional societies, museums, artists, librarians, educational amplifier, journal publishers, and curricular designer whose integration into the tournament team were described above, numerous others have volunteered, most often spontaneously, their skillsets toward enhancing the tournament. The bracket went from janky to elegant in 2016 courtesy of graphic designer Nickley, and undergraduate and graduate students have generated sports-style battle summaries that are posted across social media platforms since 2018 [Lesciotto, Krell, Martin]. Fossil ornithologist, Chen, tracks taxonomic representation and generates a color-coded combatant phylogeny annually. The Aldo Leopold Foundation provided an intermission message, sharing an enduring ethos of land stewardship through paired images and quotations from 2016 to 2019 [Kobylecky]. Launched independently via YouTube, MC Marmot and the Rodent Roundtable is a sports-style rundown puppet show that was an instant hit with school children in 2017 [Dietrick, Easterling]. MC Marmot now collaborates actively with

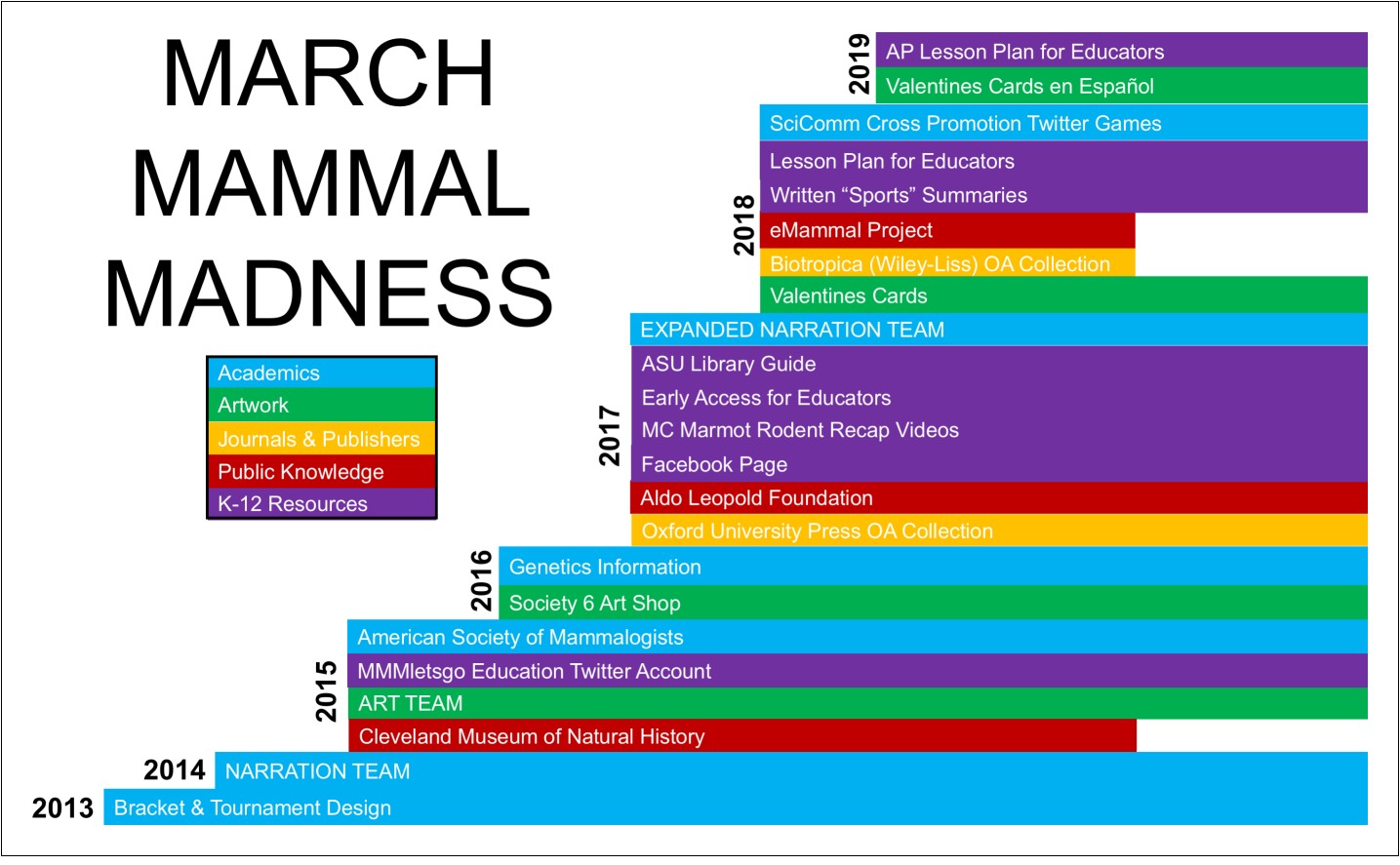

**Figure 7.** Timeline of development and new elements in March Mammal Madness. When MMM started in 2013, a single scientist-narrator designed the bracket and reported battle outcomes, but was joined by a team of scientist-narrators in 2014. In 2015, the team expanded to include artists, museum staff, and a dedicated MMMletsgo Twitter account. An academic publisher curated a special MMM collection issue for the first time in 2017. In recent years, we have expanded the teaching materials for K-12 Educators.

the MMM team as they prepare their science comedy scripts. In response to an emailed request from the principal of a school in the United States serving children with hearing-impairment, MC Marmot added closed-captioning to videos in 2019. Collaboration is a key component of successful online outreach (*Bik et al., 2015*). March Mammal Madness routinely demonstrates that 'teamwork makes the dream work' but even more exemplifies the emergent, ephemeral alchemy of a creative collective brought together through their respective knowledge, complementary skills, and shared love of the natural world.

In addition to the contributions from well-established science communicators, MMM serves as an incubator for SciComm skill development and media training for trainees and faculty. The diverse skillsets among the MMM team facilitate an annual "SciComm spring training" for messaging to the public. Scientists learn to prioritize story-telling (*Neeley et al., 2020*) and accessible accuracy in science communication (*Yong, 2010*), and these techniques are more effective with audiences than the compounding obfuscation generated by pedantic attention to inaccessible precision, indecipherable jargon, and overwhelming comprehensiveness. Contributors to MMM gain visibility, a wider audience through new followers, and an expanded social media network. Additionally, contributors' study taxa and topics are intentionally showcased in the tournament. MMM contributors have been featured in media interviews, podcasts, news stories, and blogs that discuss the tournament, expanding their media experience and connections with science journalists. In this way, the broader impacts of March Mammal Madness are twofold, both in communicating science to the public and preparing scientists to publicly communicate. Moreover, the MMM contributor community supports, mentors, cheers, and cares for each other throughout the year. Informal peer-support networks are important in the

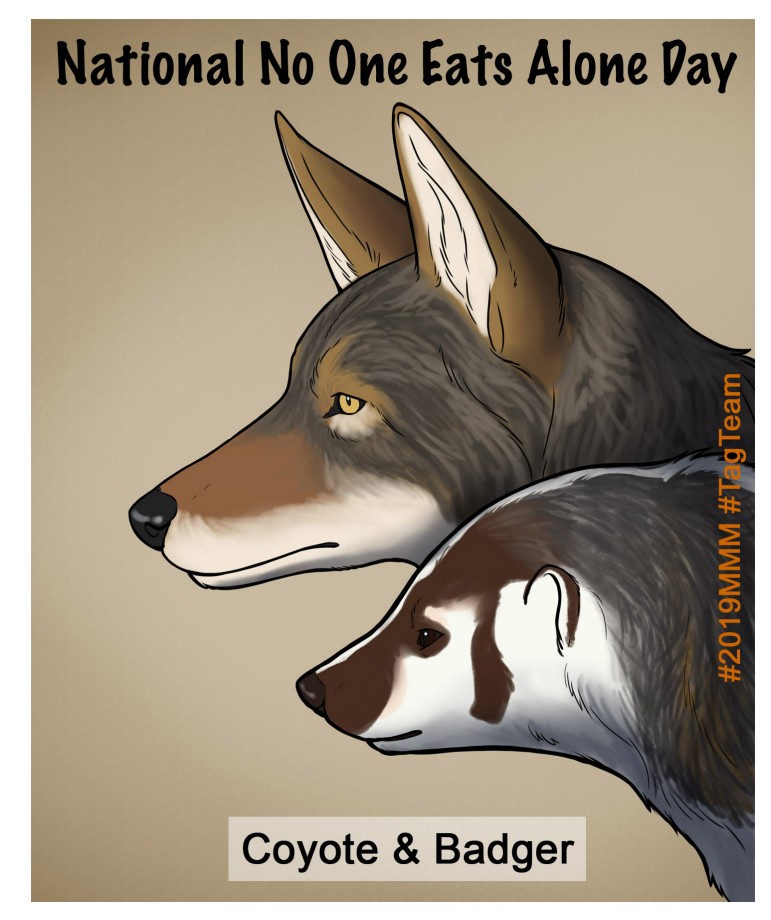

**Figure 8.** MMM promoted *National No One Eats Alone Day* in 2019. *"Today is National No One Eats Alone Day to promote inclusion and acceptance in schools!* https://nooneeatsalone.org *Did you know that sometimes Coyotes and Badgers hunt together? Coyote and Badger agree: #NoOneEatsAlone art by @Opellisms #2019MMM #TagTeam"* — @Mammals_Suck.

development of early-career researchers (*Macoun and Miller, 2014*), particularly for identities underrepresented in academia (*Agosto et al., 2016*). The use of Twitter as a primary platform expands the opportunities for informal mentoring and support and can accommodate the unfortunately transient aspects of early career stages by facilitating access to colleagues and confidants regardless of geographic location (*Ferguson and Wheat, 2015*).

## Emergent community: public, scientists, and institutions

Although tournament content is widely available across multiple social media and website platforms, the most dynamical interactive aspects occur on Twitter. Twitter not only provides the figurative amphitheater allowing spectators to actively engage during the "battles," but facilitates an active, interconnected community among the citizenry. Students, fans, scientists, academics, and institutions hilariously interact during the weeks of the tournament and, to a lesser extent, throughout the year. In this way, March Mammal Madness reaches many "publics" and explicitly dismantles boundaries among scientists, students, and the broader members of society (*Varner, 2014*; *Jarreau et al., 2019*; *Cheplygina et al., 2020*), an important component in stemming misinformation (*Scheufele and Krause, 2019*).

Particularly compelling jokes, combatants, themes, and controversies become ongoing hashtags (*Buarki and Alkhateeb, 2018*). Hashtags, such as #2019MMM, function to coordinate creators and consumers toward relevant content on social media platforms. In this way, searching or following hashtags facilitates access to topics and communities. On Twitter, users have "real time" content in their "timeline", and can use hashtags to filter popular or recent tweets. Scientist-narrator celebration of carnivore dentition has perpetuated into the perennial exclamation of #carnassials. Bloodthirsty spectators disappointed in accurate withdrawal outcomes have for years hollered for #carnage. In response, plant biologists now routinely decry the rampant #PlantCarnage perpetrated by herbivores in battle narrations. In 2016, the giant panda was described as simultaneously "the worst bear" and "the worst herbivore," due to poor digestion of the cellulose that comprises the majority of the panda's diet – earning the continuing moniker #WorstBear (*Woolston, 2016*). In 2019, the inclusion of mutualists Bornean Bat (*Kerivoula hardwickii*) & Pitcher Plant (*Nepenthes hemsleyana*) not only inspired the hashtag #TeamBatToilet, but also the fan-created Twitter account @TeamBatToilet that heckled, cheered, and informed throughout the tournament. One particularly purrsistent fan-generated hashtag has been #CatScandal, as felid aficionados pawsited that systematic bias, rather than infurriority, contributed to the early exits of cat combatants from the tournament (*Kosmala, 2016*).

But one MMM joke outsizes them all (no, not the *Paraceratherium* 'Walter'). During a 2016 first round mustelid-e-mustelid battle, Prof. Kristi Lewton narrated the relative mass "1 wolverine = 67 stoats," a hilarious device subsequently applied to additional battle narrations as numerous combatants were converted into stoat units. Several nights later, Lewton reported her

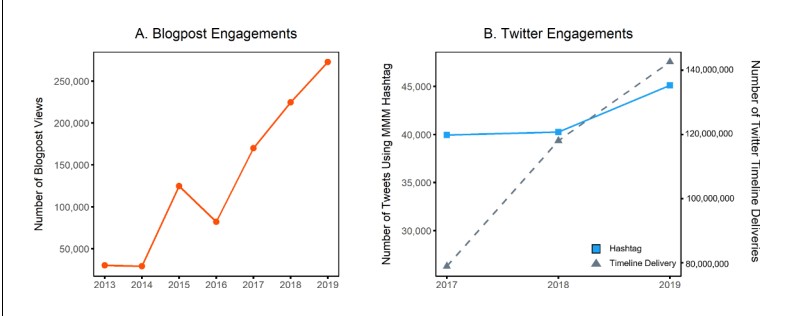

**Figure 9.** Increasing engagement on social media. (A) The number of pageviews for MMM blog posts increased over time, as did engagement on twitter (B), as measured by the number of tweets using the MMM hashtag (solid blue line) and the number of timeline deliveries (dashed grey line).

discovery that the stoat unit of measurement was used as early as 1866 when esteemed natural historian and Royal Society Fellow George Allman described an otter shrew as "somewhat larger than a stoat" in his treatise on the clade in the Transactions of the Zoological Society of London. Subsequently the artistic director and editor collaborated to create an official conversion chart. To date, #StoatsAsMeasurement remains one of the most popular MMM hashtags among fans (and scientist-narrators), routinely tweeted hundreds of times each year.

March Mammal Madness intentionally builds connections with other science communication and education campaigns. Battle narrations routinely use well-established science Twitter hashtags such as #ActualLivingScientist #MammalWatching, #UnderratedUngulate, #PoopScience, and #FieldWorkFail (*Becker, 2017*, *Feldkamp, 2017*, *Irwin, 2018*; *Jourdane, 2017*) that have crossed-over into mainstream media discourse. To launch the MMM "preseason" the first week of February beginning in 2018, we collaborated with established twitter games #CougarOrNot, #StreetCreatures, #GuessThatCrest, #TrickyBirdID #NameThatMammal #ButtOfWhat and #NameThatCarcass, helmed by experts in mammalogy, ornithology, and urban animals (*Bartels, 2017*; *Becker, 2019*; *LaRue, 2018*) for a SciComm cross-promotion extravaganza of MMM combatant reveals. In recent years, museums have engaged in tongue-in-cheek twitter flame wars to showcase their collections, giving rise to #MuseumSnowBallFight (*Nied, 2018*) and 'Best Duck' (*Birkhead, 2019*). In 2018, the American Museum of Natural History defeated the Field Museum in their MMM bracket competition. This museum bracket challenge expanded in

2019 to eight museums, but the AMNH's champion *Nimravid* was eliminated in the 2nd round in a stunning upset that featured scientific findings from the AMNH's own archives (*Toohey, 1959*). The museum Twitter accounts provided light-hearted and hilarious interactions, thereby bringing #2019MMM to their social media communities. The Tag Team Division of species mutualisms in 2019 presented an exceptional opportunity to highlight National No One Eats Alone Day on February 15th, a student-led effort to promote social inclusion and acceptance (*Figure 8*).

User engagement in the March Mammal Madness tournament increased across multiple domains and platforms over the years. Views of the annual tournament blogpost have increased ninefold from N = 30,000 in 2013 to N = 272,000 in 2019 (*Figure 9A*) a rate of growth exceeding the background growth in Twitter (*Leetaru, 2019*). We tracked hashtag use on Twitter during the 2017–2019 tournaments. Although ~1400 tweets annually are official tweets generated by the MMM team, an additional 40,000+ tweets are created or shared by the active MMM Twitter community (*Figure 9B*). In 2019, the highest annual hashtag use to date, 5400 accounts used the tournament hashtag, tweeting to 13.3 million followers. Cumulative estimates of timeline deliveries of tweets using the tournament hashtag 2017–2019 are in excess of 339 million, although not all tweets will be seen by all followers (*Figure 9B*). On Twitter, as of fall 2019, the tournament account had 17,000+ followers and retweeted only official tournament tweets by organizers and contributors, thus showcasing only scientific and artistic content while shielding followers from any fandom intensity that manifests as profane exclamations on the tournament hashtag. This "MMMletsgo" account was spontaneously created in 2016 by then high school junior Emma Willcocks, and she continued to maintain the account as a college undergraduate majoring in Biology. All official tournament tweets since 2013 have been archived, initially on Storify, but with the scheduled extinction of that platform in 2018, the March Mammal Madness collection was migrated to Wakelet where it continues to be curated. All scientific content of tournament battles remains available and, to date, the archive has been viewed tens of thousands of times. As of Fall 2019, 6,500+ accounts followed the March Mammal Madness Facebook page and the day the 2019 tournament bracket dropped the FB post organically reached

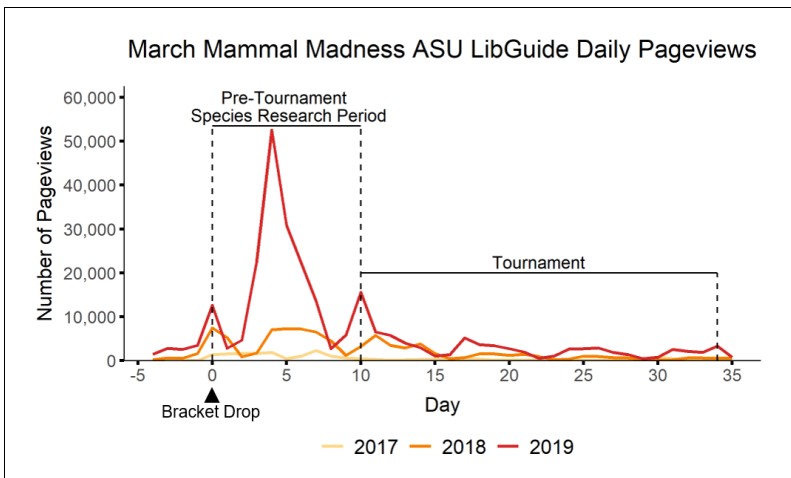

**Figure 10.** Pageviews of the ASU LibGuide before and during the MMM tournament. Daily page views for the MMM ASU LibGuide were greatest during the pre-tournament research period, but active traffic was sustained during the tournament as seen for 2017, 2018 and 2019; for each year, day 0 is the day the tournament bracket was released.

43,000+ Facebook newsfeeds from user engagement. These social media engagement numbers for followers, shares, and retweets indicate that tournament content is broadly reaching public audiences (*Côté and Darling, 2018*; *McClain, 2019*). Moreover, social media engagement around natural world content has been associated with increased donations to conservation campaigns (*Lenda et al., 2020*) and long-term changes in species awareness (*Fernández-Bellon and Kane, 2020*).

## Educational resources, propagation, and impact

Beginning in 2017, Arizona State University (ASU) Librarian Anali Perry and colleagues created a March Mammal Madness Library Guide (LibGuide) to provide links to freely available, *reliable* online sources of animal information for students and others as they make their bracket predictions (*Perry et al., 2017*). LibGuides are a standard platform to provide information, collect resources, and curate content around a theme or subject and are the primary proprietary guide-creation platform within library sciences (*Bowen, 2014*; *Griffin and Taylor, 2018*). Developed by Springshare in 2007, LibGuides are designed to be easy to create and update directly by library staff, like a blog interface, and structured for intuitive navigation by users (*Bowen, 2014*). The platform collects usage statistics and can generate customized usage reports to assess how users are navigating the resource (*Gessner et al., 2015*; *Griffin and*

*Taylor, 2018*). Across tournament years, use of the ASU Library March Mammal Madness LibGuide has increased 14-fold, from N = 18,992 page views in 2017 to N = 274,926 in 2019. Not only is this the highest traffic LibGuide created at ASU, in 2019 the MMM LibGuide was the 125th out of over 700,000 LibGuides on Springshare, putting it in the top 0.0002% on the platform. Each year, the top three elements of the MMM LibGuide have consistently been the 'How to Play' (38 ± 7%), 'Annual Tournament Information Page' (29 ± 4%), and 'Animal Information' (26 ± 4%). The 'Animal Information' page of the LibGuide links to resources such as Animal Diversity Web, Smithsonian's National Zoo and Conservation Biology Institute, and the Encyclopedia of Life as students conduct background research to make predictions for bracket outcomes. Use of the MMM LibGuide is primarily during the pre-tournament period after brackets of species combatants have been publicly released but before the tournament battle narrations have begun (*Figure 10*). Importantly, the MMM LibGuide provides a stable location for the tournament information year-to-year to aid educator and student use and the .edu webaddress is not typically blocked by school or library public computer browser filters (*Cameron et al., 2019*).

Oxford University Press has curated a special issue of articles from the Journal of Mammalogy and Mammalian Species that feature combatant species since 2017. This special issue is hosted under the OUP banner of the American Society of Mammalogists Journals. Initially providing nine articles to the top-seeded combatants in each division for 2017, the special issue has expanded to include articles for N = 20 mammalian species in 2018 and N = 25 in 2019. Traffic to the special issue each March has been monotonically increasing from N = 1743 pageviews in 2017 to N = 12,110 in 2019. Indeed, in 2019, traffic to the March Mammal Madness special issue accounted for over 14% of all traffic to the journal for the entire month of March.

Educators have increasingly adopted March Mammal Madness due to word-of-mouth about teacher and student enthusiasm, intentional design of curricular materials, and educational resources such as the ASU LibGuide. In response to informal teacher feedback, we invited educators in February 2017 to submit requests for early access to the bracket to facilitate planning for classroom use before it became publicly available. We expanded this practice in 2018 to include not only early release of the bracket, but

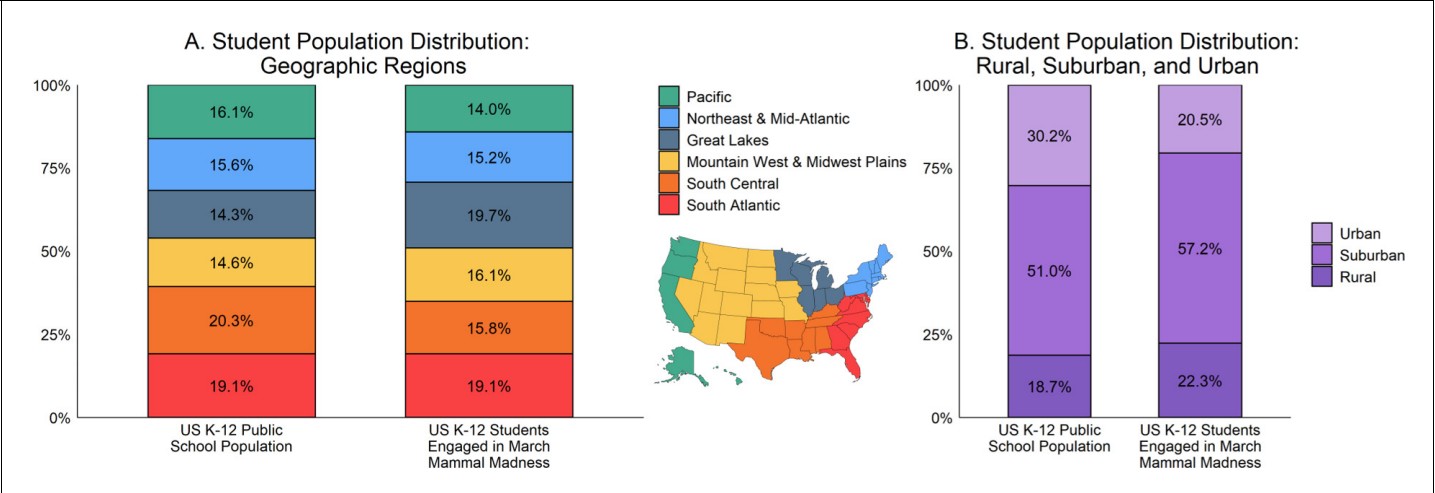

**Figure 11.** Interest in MMM by schools across the United States in 2018. (A) The proportion of the total public school K-12 student population in six geographic regions (left) and the proportion of MMM students in these regions (right); the two distributions are largely similar, but involvement in MMM is proportionately lower in the South Central region and higher in the Great Lakes region. (B). MMM was under-represented among urban communities and over-represented among suburban communities.

pre-tournament and tournament lesson plans and worksheets for educators to integrate MMM into their science classrooms (see *Supplementary files 4* and *5*). The lesson plan included a pre-tournament research phase in which students chose (or were assigned) 1–2 of the 65 animals in the tournament bracket. Students then created animal profiles from researching the animals' biomes, adaptations, and trophic levels. Once each annual tournament began and scientist-narrators provided narrative play-by-plays explaining the battle outcomes, students completed worksheets comparing and contrasting their predictions with the scientific explanations from the official tournament outcomes. The lesson plans and worksheets prompt students to answer questions about the species relating to Next Generation Science Standards: behavior, evolution, adaptation, human impacts, and ecosystems (*National Research Council, 2015*). Beginning in 2019, we developed additional permutations of the worksheets that emphasized anatomy and physiology, classification system, and genetics, partly in response to survey findings from 2018 (described below) that revealed the breadth of courses taught by educators using March Mammal Madness. Additionally, as few Americans can name a living scientist (*Research!America, 2020*), the worksheets prompted students to report information about the scientist(s) who conducted the research that was cited in the battle. To better harmonize tournament content with classroom curriculum, internal MMM

protocols for battle narrations were updated annually to coordinate battle narration content with the student worksheets distributed to educators. In this way, we have positioned March Mammal Madness for propagation and sustainable adoption by educators (*Stanford et al., 2017*).

Sequential surveys of educators in 2018 and 2019 indicate that March Mammal Madness has been adopted across all continents except Antarctica, reaching hundreds of thousands of students since 2013. The 2018 survey prioritized a quantitative assessment of the educational contexts in which educators were distributing the tournament bracket to students, whereas in 2019 we conducted a more qualitative assessment of how educators were using the tournament in their classrooms and their perceptions of student impact. Among educators requesting March Mammal Madness open educational resources in 2018 and 2019, an astonishing 99.6% and 99.7% opted to participate in the annual survey, although not all respondents answered each survey question (for information about surveying educators and more typical response rates of 20–30%, see *Neal et al., 2020*). In 2018, N = 1594 survey respondents provided information about the number of students to whom they intended to distribute the bracket (N = 119,768 students), courses and grade levels they taught, and the rural/suburban/urban context of their school and its geographical region. In 2019, N = 3171 survey respondents requested March Mammal Madness

materials to use with their N = 245,483 students and provided information about how they found out about the tournament and whether/how they would integrate these materials into their curriculum. We note that 37% (N = 1173/3162) of the educators responding to the 2019 survey had previously used March Mammal Madness in their classrooms and may have continued to teach some of the same students, so we are unable to definitively combine the student totals across 2018 and 2019 to generate a cumulative number of students. Regardless, we expect that these educator and student numbers likely underestimate the reach of the tournament because we release the bracket and teaching resources from an embargo over a week before the tournament begins. At that point, the bracket and teaching resources become freely available and are likely widely shared within and across educator groups and websites. Indeed, in the 2019 survey, educators reported they were most likely to have found out about the tournament through Facebook teacher groups (N = 1360/3157; 43%) or directly from colleagues (N = 674/3157; 21%).

The majority of educators using March Mammal Madness teach life sciences to high school students and are proportionately distributed across the United States. In the 2018 survey, nearly all educators were situated within the United States (N = 1538/1593, 96.5%) as were their students (N = 117,079/119,745 students; 97.7%). Over ninety percent of the educators using March Mammal Madness taught classes in the life and earth sciences (N = 1448/1586; 91.2%), particularly biology and/or environmental science (N = 1093), but zoology, anatomy and physiology, geology, oceanography, mammalogy, ecology and evolution, zoology, and other sciences were represented. Educators outside the life sciences taught general education, humanities, math/statistics, physical sciences, special education, science communication and other courses. March Mammal Madness is primarily used by K-12 teachers (N = 1516/1589, 95.4%), mainly high school (grades 9–12; N = 1099) and middle school teachers (grades 6–8; N = 244). A smaller proportion of the respondents were elementary school teachers (K-5; N = 80) and college faculty (N = 72), or taught across elementary, middle school, and high school boundaries (N = 94). Importantly, datasets made available through the National Center for Education Statistics from the U.S. Department of Education allow us to evaluate MMM reach within the broader context of education in the United States (*Glander, 2017*). March Mammal Madness use was largely proportionately distributed across geographic regions of the United States (*Figure 11A*) based on SY15-16 (*Glander, 2017*), the most recent year for which data are available. Although over-represented among rural (N = 25,857/115,433; 22.3%) and suburban (N = 65,812/115,443; 57%) communities, and under-represented in urban communities (N = 23,714/115,443; 20.6%), in 2018 March Mammal Madness was distributed to K-12 students somewhat similarly to their distribution across urban-suburban-rural gradients in the United States (*Figure 11B*; *Glander, 2017*). Assuming consistencies with 2018 demographics, the increased participation of educators and their students in March Mammal Madness in 2019 suggests that the tournament reached ~1% of high school students in the United States (*National Center for Education Statistics, 2019*).

Even while highlighting how the tournament is fun, most educators implemented March Mammal Madness with pedagogical intention in their classrooms. In the 2019 survey, educators reported that they most typically planned to use the tournament as an embedded component in units on adaptation, diversity of life, biological interactions, human impact, ecosystems, taxonomy and other topics to introduce, discuss, reinforce, or review course content (N = 2119/3026, 70%). Over a quarter of educators planned for students to engage in the tournament through in class activities often involving a combination of pre-tournament research, presentation, and/or project (individual or group) to support critical thinking, team-building, and 'explain, justify, argue from evidence' skills (N = 852/3026, 28%). Very few educators planned to only use the tournament for an extra credit activity (N = 53/3026, 1.8%). Educators who had familiarity with the tournament prior to 2019 were more likely to explain how the tournament would be implemented with a specific plan/purpose than were educators participating for the first time in 2019 (N = 1107/1136, 97% vs. N = 1359/1883, 80%; $Chi^2$ = 224.3, p<0.0001). In many cases, students would present their background research on an animal combatant through a promotional poster or public speaking. Relatively few educators integrated art, creative writing, or group work in conjunction with March Mammal Madness in 2019. Numerous teachers described building a large bracket in school hallways surrounded by student-generated, species summaries:

*"Students will research animals and adaptations and write a paragraph about why their animal could win MMM. They will then create some sort of artistic representation of the animals. Students will then participate in a gallery walk in order to help them complete their bracket."* —Educator Respondent

Educators reported that March Mammal Madness is emotionally and intellectually engaging for their students. In both annual surveys, the final prompt was an invitation for the educators to share any comments they had about the tournament. In 2018 and 2019, ~90% of educators who responded to this prompt included positive content (N = 265/279 and N = 632/704, respectively) with fewer than 4% of comments including negative content. Semantic textual analysis (*Bree and Gallagher, 2016*; *Maguire and Delahunt, 2017*) showed that 28% (N = 257/910) of educators spontaneously described March Mammal Madness as "fun," "great," and/or "awesome." Over 40% of responding educators (N = 373/910) used the word "love" – their students' love and/or their own – for March Mammal Madness. Qualitative thematic analysis with latent evaluation of educator's answers (*Bree and Gallagher, 2016*; *Maguire and Delahunt, 2017*) revealed not only the educators' appreciation that the tournament connected to curricula, but several compelling themes were identified about how the tournament stimulated emotional engagement, skill development, and interest in science. Here we include illustrative quotes from educator responses. Educators appreciated how the tournament was scientifically grounded and reinforced lessons from the curriculum.

*"I love how this activity takes into account the animals' unique physical adaptations, but their behavior (yes, the sloth broke my heart last year) as well as the biome in which the 'battle' takes place. It makes learning fun for the students AND the teachers! As a bonus, the timing is good since we've just finished studying evolution (including phylogeny) as well as ecology in AP Bio. Thank you VERY MUCH!"* —Educator Respondent

*"My students loved it and it allowed me to organically incorporate a lot of evolution and ecology that made sense because the students had a context."* —Educator Respondent

*"As a part of a self-contained class for high school students with moderate cognitive disabilities. Besides being generally informative and entertaining, it allows my students to develop functional skills such as critical thinking, making choices, organizing systems and forecasting events."* —Educator Respondent

During the tournament, students became deeply invested in their research of the animals. Educators reported students animatedly discussing adaptations and habitats with fellow students and teachers, even outside the classroom.

*"The students loved researching different organisms that they didn't know about and having arguments and discussions about the results as they came out. I had a huge bracket printed on my door and students and teachers all over the school stopped by to see and talk about results. It was very fun. One of the highlights of the school year."* —Educator Respondent

*"<Students> were so engaged in the process of filling out brackets and arguing over battle outcomes- I've never seen an activity get kids so passionate about discussing animals!"* —Educator Respondent

*"My students loved it. There were many conversations between the kids as to who will win each battle with well thought out rationale behind it and in some instances, kids stopped what they were doing to look up details about the organisms in the middle of discussion to go over more nuanced specifics about their organisms."* —Educator Respondent

*"My students LOVED it! . . . They were talking about it in the halls, at lunch. It was EPIC! I can't wait to do it again."* —Educator Respondent

This enthusiasm was sustained long-term. Students continued to discuss combatant animals after the conclusion of the tournament. Upon returning to school the next academic year, students sought verification that the class would once again participate in March Mammal Madness. Additionally, educators reported that former students, even those who have graduated from the school, would return to get the tournament bracket.

*"So engaging- kids loved it and did so much research. They still talk about it a*

*year later. I have kids that are planning to come back to my room this year for a bracket- even though they aren't in my classes!"* —Educator Respondent

*"MMM totally changed a sedate class into a group of obsessed animal lovers!*
*they can come back to fill out a bracket. Well, of course you can!"* —Educator Respondent

A small number of educators highlighted that the tournament was engaging to students who were not typically participatory in science class.

*"I was very excited when some of my least engaged students became very interested in the results and started to participate in the class discussions about MMM."* —Educator Respondent

*My kids loved it and learned a lot. I had students sign up for college biology just because they heard about MMM.* —Educator Respondent

Educators emphasized how the tournament amplified the student's energy and enthusiasm in class and that the humor and battle narration made both science and scientists more accessible to the students.

*"...Students would come to class chanting "March Mammal Madness" everyday!"* —Educator Respondent

*"I really appreciate all of the resources (aka journal articles) that connect to the topics we study in our biology class, and how the Twitter posts are both entertaining and lighthearted, as well as informational and educational. I also love being able to show "real scientists" to my students - thank you for all of the work that goes into this; my students absolutely love it!"* —Educator Respondent

Although survey responses were enthusiastic, our educator surveys have several notable limitations including selection bias, indirect access to student experiences, and unclear learning outcomes. By conducting the surveys in the lead-up to the tournament, our educator respondents represent two distinct categories: (1) educators experienced with March Mammal Madness whose positive or beneficial experiences in the past motivate sustained adoption of the tournament and (2) educators who plan to use the tournament for the first time. This design does not allow us to learn about the experiences and

perspectives of educators who, having tried the tournament once, do not sustainably adopt March Mammal Madness. Additionally, by asking about experiences one and more years ago in an online survey, recall bias may influence responses (*Bell et al., 2019*). For further research, a combined pre-tournament and post-tournament survey design and/or a smartphone survey app throughout the tournament has the potential to better assess myriad educator experiences while using the tournament with their learners. Moreover, although educators are reliable in assessing the achievement of their students (*Rimfeld et al., 2019*), educator responses to our surveys represent pooled observations and an aggregate assessment of their students' engagement with March Mammal Madness. Future research should more directly assess individual student perceptions, emotional affect, learning, and meta-cognitive outcomes as a function of participation in the March Mammal Madness tournament across time (*Jensen et al., 2017*).

## Narrative facilitates learning

The bracket-based tournament structure of March Mammal Madness functions as a narrative arc and immerses "learners in a captivating world populated by intriguing characters" (*Mott et al., 1999*). Through narrative, learners are transported across time and space, draw inferences, and experience emotions (*Gerrig, 1993*). Information constructed in narrative is easier to comprehend, read faster, better recalled and inconsistencies are more readily detected than are other forms of exposition (*Dahlstrom, 2014*; *Glaser et al., 2009*). Narrative-centered learning has important motivational benefits by promoting learner self-efficacy, interest, presence, and perception of control (*McQuiggan et al., 2008*). Moreover, narrative-based educational activities enhance learning and memory by working within cultural frameworks and cognitive architecture (*Mott et al., 1999*; *Neeley et al., 2020*). Due to computational demands of content processing, the effectiveness of narrative-based education is contingent on scientific information being integral to the story (*Fisch, 2000*). Instead of sharing lists of animal facts or relegating outcomes to a process of voting, March Mammal Madness scientist-narrators present facts embedded in suspenseful descriptions of combatant's offensive and defensive maneuvers as though observing such an encounter in real time. In this

heightened, shared moment, we are all as naturalists observing animal behavior, imagined in the mind's eye. The dynamism of narrative enhances emotional engagement among players (*Glaser et al., 2009*), especially elements of suspense (*Gerrig, 1993*).

> *"Oh, right, something we forgot to mention until JUST RIGHT NOW... that might be important... Since it's early spring, our bull moose is of course without antlers, having dropped them back in winter as all deer species do. #2019MMM"* —Scientist-Narrator Tweet

Narratives engage mental models – constructs of character traits and goals within the rules of the "story world" – within the audience (*Glaser et al., 2009*; *Gerrig, 1993*). Notably players are adept at recognizing that in this manufactured March Mammal Madness story world, they are "spectating" on naturalistically-inspired encounters. The animal combatant is constructed as oblivious to any tournament and therefore can have very divergent goals and motivations from the spectators. This situation precipitates many hilarious Twitter exclamations of encouragement, especially when considered through the multiple layers of imagination and theory of mind. Since the play-by-play is written in advance, but the announcing "occurs" as though in real-time on social media, effectively the spectator is yelling at a representation of an animal in their mind, collaboratively crafted by their pre-existing knowledge and the information being provided by the scientist-narrator (*Gerrig, 1993*). In this way, storytelling represents iterative theory of mind among narrators and audiences (*Bietti et al., 2019*).

Importantly, the gamified bracket format "story arc" facilitates exploration, collaboration, and reflection among students (*Mott et al., 1999*). Presenting a list of 60+ animal species and tasking students with researching their adaptations and ecosystems would likely manifest as onerous busy-work, but gamification of those same species arranged in a bracket with the question "Who Would Win?" skyrockets student psychological and emotional engagement (*Hamari et al., 2014*; *Lee and Hammer, 2011*; *Subhash and Cudney, 2018*). Educators routinely highlight the collaborative discussions among students during pre-season research, as they speculate and hypothesize about various attributes, environments, and other contingencies that may influence the tournament outcomes. Educators reported that the tournament

facilitated assignments on conducting research, critical thinking, and generating reasoned claims from evidence (*McNeill and Martin, 2011*). Importantly, during in-person learning, nearly 100% of US-based schools have internet access in classrooms, computer labs, or a school library to facilitate their research of combatant taxa (*Fortner et al., 2018*). In conjunction with discussions among classmates, students individually generate predictions of the outcomes of combatant encounters across tournament rounds until they construct a completed bracket and identify their tournament champion. In this way, students are active agents in their learning (*Reeve and Tseng, 2011*) and co-constructors of narratives (*Mott et al., 1999*), creatively integrating animal and ecological information in new combinations across tournament rounds. March Mammal Madness, depending on how the tournament is delivered to and perceived by learners, has the potential to access numerous dimensions underlying learner engagement. Importantly, learner engagement reflects emotional, behavioral, and cognitive investment, with personal agency and social embeddedness also playing key roles, and contributes in part to learning outcomes (*Ciric and Jovanovic, 2016*; *Veiga, 2016*).

Scientist-narrators expect students have conducted scouting research and provide added value by crafting narrative explanations for outcomes gleaned from primary literature. These outcomes may be consistent with the student's hypothesized battle or share exciting new information. As such, the March Mammal Madness format explicitly rejects the deficit-based approaches that are ineffective for science outreach (*Varner, 2014*, *Yuan et al., 2019*) and adheres to the known-new construct that effectively scaffolds knowledge and supports learning (*Mukherjee, 2018*). Further, the tournament manifests the learning environment advocated by Mott and colleagues in 1999 "...*by enabling learners to be co-constructors of narratives, narrative-centered learning environments can promote the deep, connection-building meaning-making activities that define constructivist learning*" (pg. 78)."

While educators in many subject areas, such as history and the language arts, embraced narrative-centered learning in the 20th Century, this educational device has achieved lower penetrance in the sciences (*Klassen, 2006*, *Glaser et al., 2009*). When present in science education, narrative-based approaches are often embedded within computer games, artificial

intelligence, and virtual-reality based systems (*McQuiggan et al., 2008*; *Qian and Clark, 2016*), access to which is inequitably distributed in the US and globally (*Resta and Laferrière, 2015*; *Fortner et al., 2018*). In contrast, users of March Mammal Madness can retain, reuse, revise, remix, and redistribute the tournament bracket and lesson plans at no cost to educators, students, and the general public (*Wiley et al., 2014*). Importantly, in a head-to-head match-up, a narrative-based approach without digital technology performed as well, if not better, than did an educational computer game in shaping student learning outcomes and interest in biology (*Sadler et al., 2015*).

The scientific illustrations embedded in March Mammal Madness parallels expanding initiatives for arts-integrated science instruction. Humanities and arts educational elements, integrated within STEM, are thought to better support student creativity, learning, collaboration, and enthusiasm for the life and physical sciences (*Perignat and Katz-Buonincontro, 2019*; *Kim et al., 2019*; *Hardiman et al., 2019*). A randomized, sequentially counterbalanced educational study among N = 350 5th graders in urban Atlanta, demonstrated that long-term science content retention was enhanced by arts-integrated instruction for students at basic reading levels (*Hardiman et al., 2019*). In this way integrating artistic creativity into science classrooms can contribute to addressing achievement gaps (*Hardiman et al., 2019*). Drawing organisms and observed phenomena in field journals was essential within the naturalist skillset and illustrators and biologists advocate for the resurrection of this arts-science integration within the natural sciences (*Merkle et al., 2020*; *Schmidly, 2005*).

Although Western education has been slow to restore narrative in science teaching, storytelling as pedagogy is found across human societies and facilitates intergenerational transfer of ecological knowledge (*Scalise Sugiyama, 2017*; *da Silva and Tehrani, 2016*; *Smith et al., 2017*). In numerous traditional and Indigenous cultures, knowledge and ways of knowing are intrinsically embedded in nature and children socially learn via storytelling by Elders (*Little Bear, 2009*; *Hare, 2012*; *Medin and Bang, 2014*). Oral tradition is foundational for sharing essential information about the natural world composed of numerous interconnections and relationships among entities, seasons, and land (*Little Bear, 2009*; *Eder, 2007*; *Holmes and Jampijinpa, 2013*). Among First Nations communities in

Canada "children engaged in learning that was experiential, land based, narrative and inter-generational" better situated their learning outcomes (*Hare, 2012*). In re-centering traditional knowledge and ways of knowing, Kaupapa Māori theory and practice in Aotearoa (New Zealand) make use of traditional pedagogical story-telling, and a wide family of story-tellers, for learners (*Lee, 2009*; *Smith, 2000*). Analyses of children's books revealed that books by Native American authors and illustrators were more likely to be characterized as close-up views of animals than were children's books by non-Native authors and illustrators (*Medin and Bang, 2014*). Further, decolonizing narratives of "*nature–culture relations*" and land dynamism can importantly contribute to global dialogues about the climate crisis and improve climate education (*McGinty and Bang, 2016*; *Greene, 2020*). Indeed, for many Native American, Aboriginal Australian, and other Indigenous cultures, knowledge about the interconnectedness of ecosystems, including humans, anchors constructs of land stewardship, community relations, ecological kinship, and shared health and well-being (*Medin and Bang, 2014*; *Holmes and Jampijinpa, 2013*; *Greene, 2020*).

## Human adaptations at play

A tournament of animals presented in narrative form by expert scientists is exceptionally, if not uniquely, salient for learners, especially young learners. Rigorous psychological research has demonstrated that children have content learning biases for animals, particularly dangerous animals (*Barrett, 2015*; *Broesch et al., 2014*), and even plants (*Wertz, 2019*). Additionally, children engage in ecological reasoning, referring to habitat relations when presented with pictures of biological species, though cultural differences likely shape children's spontaneous reasoning about food chain relations and biological needs (*Medin and Bang, 2014*). Notably, humans are characterized by a particularly extended period of juvenility (*Crittenden and Meehan, 2016*) that involves substantial social learning via story-telling, a pedagogical approach disrupted in Western schooling practices (*Scalise Sugiyama, 2017*; *Neeley et al., 2020*). Cross-culturally, children readily attend to learning from knowledgeable individuals (reviewed in *Boyd et al., 2011*; *Kline, 2015*). Anatomical, cognitive, neurobiological, and cultural capacities for language, cooperation, and control of fire (*Sugiyama, 2001*; *Smith et al.,*

2017) afforded human social groups extended hours for a "virtual world of the imagination, ritual and stories" (*Wiessner, 2014*). Indeed, across numerous cultures end-of-day fireside gathering of family and friends is often dedicated to story-telling (*Wiessner, 2014*; *Smith et al., 2017*). Animals feature prominently in many oral traditions, stories, and folklore and may represent fitness-relevant information for predator avoidance, hunting success, and safe navigation (*Sugiyama, 2001*; *da Silva and Tehrani, 2016*). These evolved capacities for content biases, storytelling, and social learning reveal that humans are adapted for narratives about the world we navigate.

Additionally, for tens of thousands of years, human creativity has manifested in artistic representations of animals. From the 35,000 years-old cave painting of a babirusa in Sulawesi, Indonesia (*Aubert et al., 2019*) to the depictions of extinct marsupial megafauna *Thylacoleo carnifex* by Aboriginal Australians (*Akerman and Willing, 2009*), human artists have exquisitely portrayed the physical and behavioral traits of sympatric species. Such artwork reveals essential natural history knowledge. For example, petroglyphs featuring predator-prey dynamics, often between felids and cervids, are found among the Scythian nomadic Iron Age culture of the Altai mountain region (*Fitzhugh, 2009*). The behavioral attributes of life history stage are shown in the hiding young steenbok and following elephant calf in the rock paintings in South Africa (*Parkington, 2003*). Moreover, animal depictions in Paleolithic cave art correlated with faunal availability in the local ecology and likely reflected necessary knowledge for successful hunting (*Rice and Paterson, 1986*). Animal motifs are found widely adorning the architecture of antiquity such as the lions on the Ishtar Gate of Babylon (*Rodler et al., 2019*) and the jaguars on Olmec monuments in the Americas (*Grove, 1972*). These animal depictions can range from realistically zoomorphic to the abstractly symbolic. In more recent centuries, scientific illustration, clay or glass models, and taxidermy became common approaches to making life-like the animal kingdom (*Péquignot, 2006*; *Topper, 1996*). Within this human tradition, March Mammal Madness has been greatly enhanced by the ongoing contributions of an incredible artistic team (*Figure 5*). Indeed, through illustration and narrative, these stories of science are crafted, and made indelible in our 'hearts' and minds.

March Mammal Madness narratives provide a collective spectator experience that emerges from multiple dimensions of human psychology and cognition. The real-time, single elimination tournament structure manifests a virtual "event" in which participation can vary along a continuous spectrum (*Getz and Page, 2016*; *Davies, 2019*; *Yoshida et al., 2014*) from minimal research in bracket selections to deep immersion in every battle. To the extent that an individual participates and engages with others, the event manifests as a dynamic, community-building experience that motivates repeat participation (*Getz and Page, 2016*; *Jahn et al., 2018*). The emergent "communitas – a temporary sense of closeness and camaraderie" among participants (*Jahn et al., 2018*) likely contributes to enthusiasm for March Mammal Madness even when one's selected champion is defeated in a battle narration (*Yoshida et al., 2014*). Players routinely tweet about deep emotional engagement as scientist-narrators tweet the battle play-by-play, describing their own shouting, cheering, laughing, jumping, and yelling in response to animal maneuvers and battle events (and the startled responses of their families, roommates, and pets in response to exclamations). Players have even expressed bewilderment at their own emotional investment in an imaginary tournament as they find themselves choked up about the fictional death of a beloved combatant. Educators described friendly competitions among their classes, school-wide engagement, and, in one case, a cross-town rivalry. Educators have also offered extra credit, trophies, or merely bragging rights for "Beat the Teacher" and "Beat the Principal." The many unfamiliar species and the secrecy of the battle outcomes "evens the playing field" between educators and learners (for once teachers DON'T already know the answers!), and among learners, between high-achieving students and their classmates. This "leveled play" aspect of the tournament likely facilitates wider buy-in among learners. The game mechanic elements within the tournament structure are combined with gamified rewards as implemented in classrooms and among social groups of co-workers, friends, and families in the forms of points, trophies, and prizes. Gamified learning often improves learner attitude, engagement, and performance, but research on gamification and game-based learning has been primarily conducted among college students (*Subhash and Cudney, 2018*).

Although the March Mammal Madness tournament is finite in duration each year, the

resonating emotions, enduring communities, and retained knowledge suggest a lasting impact. Past tournament events are routinely revisited through hashtags and retelling of stories. Such activities contribute to the formation and maintenance of a collective tournament memory and group history (*Bietti et al., 2019*). Interactions with nature and live animals can build enduring connections with the general public (*Bush et al., 2018*; *Schuttler et al., 2018*) but present ethical, logistical, scalable, and safety challenges in many contexts. We speculate that some of the animal "characters" that emerge from MMM story arcs make similar, lasting connections, without commensurate costs to a living animal and partially bridge the loss of human-nature interactions in increasingly urbanizing human populations. Parents have emailed hilarious photos and stories of their children at zoo exhibits of species featured as MMM combatants. Moreover, although we routinely select cute, familiar, and dangerous mammals that appeal to content biases among children and adults, the inclusion of rare taxa and their ecosystems raises their visibility and familiarity for hundreds of thousands of students and the general public. By weaving together elements of the humanities and social sciences into the tournament, both in the delivery and design, March Mammal Madness models important approaches to science communication (*Bush et al., 2018*; *Neeley et al., 2020*), scientific literacy (*Roth and Lee, 2002*), and biodiversity conservation (*Bennett et al., 2017*; *Lenda et al., 2020*). Importantly, by crafting stories of organisms and the rich details of their lives, and highlighting the exquisite work of well-known and emerging naturalists, March Mammal Madness contributes to a necessary "revitalization of natural history" (*Tewksbury et al., 2014*) that fosters curiosity-driven learning (*Farris, 2020*).

March Mammal Madness is widely appealing and facilitates myriad connections among numerous publics. The combination of animals, bracket, experts, and narrative absorbs diverse audiences across geographic regions, rural-urban gradients, and age groups. As "Nerds of Trust" (*McClain, 2017*), we have fielded queries from grandparents, afterwork drinking buddies, hospital radiographers, retirees, Hollywood industry workers, veterinarians, high school students, and many others. Educators report the enduring enthusiasm of their students, including students not typically engaged in the science classroom. As such, March Mammal Madness reaches beyond typical SciComm audiences with established interests in science (*Ocobock and Hawley, 2020*). The tournament, however, also has extensive traction across university, museum, and conservation communities. Scholars have referenced the tournament in various academic publications including in the acknowledgements of a PhD dissertation (*Woods, 2018*), in a book review (*Fox, 2018*), and in an article figure description in which *Paraceratherium* is called 'Walter' from #2014MMM (*Sulak et al., 2016*). The tournament can also be effective for settling sticky scholarly situations; *Brisson-Curadeau et al., 2017* acknowledged MMM bracket score for determining author order (2017).

Multiple measures of engagement reveal that tournament participation has grown annually since 2013, reaching at least 250,000 people in 2019. To put that in an available context, the National Museum of Natural History and the Smithsonian National Zoological Park reported N = 427,421 and N = 138,676 visitors respectively in March 2019 (*Smithsonian Institution, 2019*) and the biennial USA Festival of Science estimated N = 370,000 attendees in March-April of 2018 (*Science and Team, 2018*). Few studies have assessed the long-term learning outcomes of zoo, museum, and science festival visits, as such outcomes are shaped by a constellation of factors, but such experiences for children and adults are important exposures to animals, biological systems, scientists, and self-directed exploration (*Godinez and Fernandez, 2019*; *Mujtaba et al., 2018*; *Davies, 2019*; but see *Jensen et al., 2017*). The extent to which participation in March Mammal Madness increases scientific knowledge among audiences similarly remains to be determined, but reports from educators emphasized that the tournament sustainably engaged learners and facilitated individual and collaborative practice with consolidation of information, advanced planning, and critical thinking. These are essential, broadly-applicable skills not only for science learning, but for academic development and life in general (*Gordon et al., 2009*; *Tan et al., 2017*).

> "If facts are the seeds that later produce knowledge and wisdom, then the emotions and the impressions of the senses are the fertile soil in which the seeds grow... It is more important to pave the way for the child to want to know than to put him on a diet of facts he is not ready to assimilate." —Rachel Carson, The Sense of Wonder, 1965

## Conclusion

March Mammal Madness upends the stereotype of science as dry, prescriptive disciplines and shows that science and scientists can be, *and should be*, creative and fun. Scientists situate ourselves in the domain of data collection framed by hypotheses and predictions as we speculate about the world(s) around us. But fundamentally these are just grown-up words for ideas hewn from imagination and the creative combination of what is known to journey into the unknown. March Mammal Madness is collective, "performance science" – the stories of animals, told creatively with awe for the natural world. We celebrate species and the ecosystems they inhabit, the scientists who conduct studies, and the funders who make the research possible. For a few weeks each year, a vibrant and diverse March Mammal Madness community comes together to collectively marvel at our living planet's beauty, harshness, and fragility. We acknowledge that humans are at the root of many of the problems we highlight, but also recognize that the communities we reach are essential branches of any solutions. By fostering a greater love and respect for biodiversity, we hope that engaged students and curious publics will be inspired to transform their affection into action and reverence into protection.

## Materials and methods

### Species

In our count of species combatants 2013–2019 (*Figure 2*), subspecies were not counted as unique combatants; *Papio* systematics counted as per *Jordan et al., 2018*; the batfly commensal *Gammaproteobacteria* were considered a single operational taxonomic unit; mythical combatants, though purportedly sharing features with biological species, were not counted as species. Order and class assignment of extant taxa of MMM combatants was systematized using R (*R Development Core Team, 2017*) taxize package that uses multiple sources for these taxonomic designations (*Chamberlain and Szöcs, 2013*) and were compared with reported species proportions among mammalian orders as described by *Burgin et al., 2018*.

### Usage analytics

Online platforms including Twitter, Facebook, LibGuide, and BlogSpot make freely available some analytics about the traffic or engagement with the account. For some of these, we were able to identify the total number of unique followers/users, daily and/or cumulative pageviews, and user engagement and amplification. Hash-tracking is a proprietary subscription service that collects metrics and metadata associated with social media hashtags including the number of tweets that have used the hashtag, the number of accounts using the hashtag, and the total followers of the accounts using the hashtag. The product of these measures generates a total number of deliveries of tweets with the hashtag during a period of time. Through our hashtracking account (Hashtracking, Ladera Ranch, CA, USA), each year 2017–2019, we tracked hashtag usage information from ~2 weeks before the bracket drop through until 3 days after the Championship battle (tournament dates shifted from year to year). Hashtracking also gleans information about device usage, temporal patterns, and other hashtags typically covarying with the focus hashtag.

### Educator survey and analysis

In 2018 and 2019, we launched a google form for educators to request early access to the tournament bracket, lesson plan, and worksheet materials before the bracket was publicly released on the Mammals Suck. . . Milk! blog and the ASU MMM LibGuide. We announced the education materials request form and provided a link via Twitter, Facebook, blog, and LibGuide. In the request form, educators were invited to answer a brief, IRB-approved survey after submitting their email address for materials and were informed that whether or not they participated in the survey had no bearing on access to materials, that they could answer as many or as few questions as they wished, and they could stop participation at any time. The full 2018 and 2019 survey instruments are included in as *Supplementary files 6* and *7*, respectively. Both the 2018 and 2019 surveys asked specifically how many years the educator had been using March Mammal Madness with their learners (allowing differentiation of experienced and first-time tournament users) and how many students they planned to distribute the bracket to. The 2018 survey asked open-ended questions about the courses/classes and what grade levels the educator taught, specific USA geographical region operationalized by states, or non-USA North America, Central and South America, Sub-Saharan Africa, North Africa and the Middle East, Central Asia, Australia and the Pacific Islands, South Asia and Southeast Asia, and Europe. Respondents were asked if their local

community was rural, suburban, urban (or other) without specifically operationalizing these terms (stage whisper: whoops). The 2018 survey asked how they used the tournament in their classroom. Respondents in 2018 for the last question were prompted to "*Please add any comments you wish to share about MMM.*" The 2019 survey asked an open-ended question about how educators had learned about March Mammal Madness and asked specifically "*In 2019, how will you use MMM in your classroom?*" In 2019, the final question we asked was "*If 2018 was the first year you used MMM in your classroom, please share any comments you have about the experiences of 2018.*"

Survey responses were evaluated for errors, duplicates, and outliers and then coded for analyses. From the 2018 survey we removed duplicate entries (N = 59), and excluded respondents who did not provide an email address (and therefore could potentially be duplicates; N = 9) and one student who requested materials for their math club, resulting in N = 1594 educators who participated in the survey from the 1598 who requested educational materials (response rate 99.6%). We censored one cell in 2018 that reported the tournament would be distributed to 5000 students, as this number was many multiples (5x) above the continuous distribution of responses to this question. From the 2019 survey we removed duplicate entries (N = 196), and excluded respondents who did not provide an e-mail address (and therefore could potentially be duplicates; N = 19), resulting in N = 3171 educators who participated in the survey from the N = 3184 who requested educational materials (response rate 99.7%). We censored one cell in 2019 that reported the tournament would be distributed to 3500 students, as this respondent indicated that they would distribute materials to teachers in their district to consider distributing to students.

For survey questions that were open-ended, respondent answers were systematically reviewed, binned (for example answers '7th and 8th grade' binned with 'grades 7 and 8' as Middle School; Twitter, twitter, tweet binned together). For our 2019 survey question about how the educator planned to use the tournament in their classroom, N = 3027 provided a textual answer. Answers were coded as either 'specific plan' or 'non-specific plan.' Examples of specific plans ranged from "*Research project*" to "*Students will create "profile sheets" for one of the animals, which will be displayed in the hallway for reference and passers-by educational*

purposes. Students can use these profiles to inform their bracket choices. Discussions over battles in class as time allows. Students who beat my bracket receive extra credit.*" Examples of non-specific plans included "don't know" and "not sure."

After data cleaning, and organizing, we were able to tabulate and analyze responses within survey year and, for one analysis, combine answers from both survey years. We conducted a $Chi^2$ analysis to compare the probability that an educator would provide a specific plan as a function of being a "veteran" or "newbie" user of the tournament using JMP 14 (SAS Institute). While assessing responses for the presence or absence of specific plans for using March Mammal Madness with their learners, some terms repeatedly occurred within the answers. KH used these terms to refine exploration of how educators planned to use MMM with their learners. KH screened text for curricula integration and classroom activities by scanning for keywords within individual respondent answers using an excel formula (*Bree and Gallagher, 2016*; *Maguire and Delahunt, 2017*). The category for "curricular enhancement" was based on inclusion of 'add', 'bell', 'class', 'complement', 'connect', 'content', 'curriculum', 'discuss', 'educat', 'enrich', 'explor', 'exten' 'integra' 'intro', 'learn', 'lesson', 'look up', 'module', 'reinforce', 'review', 'section' 'study', 'supplement', 'teach', 'topic', and 'unit'. The category for "skill development" was based on inclusion of 'activit', 'argu', 'assign', 'collab', 'critical', 'debate', 'EJAE', 'evaluat', 'evidence', 'explan', 'explain' 'group', 'justif', 'present', 'project', 'research', 'reason', 'team', 'think', 'predict', 'poster', 'problem-solv', and 'problem solv'. For words that had multiple derivations, we used a word root that would capture them collectively. Given this formulaic approach, the answers were secondarily screened for accidental "by-catch." For example, a formula that assigned "TRUE" to and answer along the lines of 'in our ecology unit, students will research animals and give presentations of their scouting reports of their traits to the class' would be accurate, but 'I'm researching the tournament as I consider using it in my class' would not and would be reassigned a "FALSE" designation.

To better understand veteran educators' key takeaways about their experiences using March Mammal Madness, we combined unique respondents across the 2018 and 2019 surveys who were experienced with using MMM in their classrooms. We accomplished this by pooling

veteran educators from the 2018 survey with educators in the 2019 survey whose first year using the tournament was 2018. Of the N = 1192 educators who fit these selection criteria, N = 910 (76%) provided free-write answers when prompted to share comments in the final question in both surveys. Comments were coded as "Positive," "Negative," "Constructive," "Constructive/Positive," "Mixed Positive and Negative," and "Other." Comments were coded as positive or negative depending on whether the comment expressed positive or negative sentiments about emotions, engagement, experiences and/or outcomes from using March Mammal Madness. Comments were coded as "Constructive" if the respondent made a suggestion, wishlist, request, or other constructive critique about March Mammal Madness. If respondent comment had combinations of positive, negative, and constructive elements, they were assigned the relevant combination code. Comments were coded as "Other" if they did not have positive, negative, or constructive elements and instead addressed scheduling conflicts, description of plans, mis-entered response to a different question, or other miscellaneous responses that would have required subjective inference to apply another valence code. Latent evaluation of survey responses by KH inductively revealed several themes and we then conducted semantic screening for thematic keywords within individual respondent answers (*Bree and Gallagher, 2016*; *Maguire and Delahunt, 2017*) including "love," "engage," "fun," "discuss," and "former" using cell formulas in Microsoft Excel. We curated illustrative quotations for inclusion in the manuscript. We noted substantial variance in the length and detail of the respondents free-write answers and our blunt, preliminary textual analysis could not effectively explore many elements and nuances among the answers or comprehensively manifest the rich scholarly approaches to qualitative text analysis (*Wutich et al., 2015*; *Bernard et al., 2016*).

### Data availability

Source data are publicly available in the ASU Research Data Repository at dataverse.asu.edu/dataverse/marchmammalmadness (*Hinde, 2021a*; *Hinde, 2021b*) and linked with the March Mammal Madness Open Resources Collection (*Perry and Hinde, 2020*).

### Acknowledgements

We thank Profs. Penny Bishop, Michelle Bezanson, and PJ Perry for providing valuable comments and guidance, especially during the particular challenges of the COVID-19 pandemic, that improved our manuscript. We thank ASU library staff René Tanner, Ashley Gohr, and Mimmo Bonanni for contributions to the ASU LibGuide; Maria A Nieves-Colón, Genevieve Housman, Andrew Ozga, Tanvi Honap, Pooja Narang and Heini Natri for help to the genetics team; Cyn Rudzis, Allen McFadden, Shannon Freed, Kim Ewell, and Cas Loll for illustrations especially for the Were-Yeti, and John Doty for the "official" Unofficial scoring sheet. Thank you to Sam Hemenway and the production team at the Journal of Mammalogy for providing special issue pageview information and Ginna Nicolas for relative web traffic on Springshare. Special thanks to the American Society of Mammalogists Informatics Committee and chair Sean Maher for supporting the use of the image collection and official ASM account for MMM tweets. We so greatly appreciate Michelle LaRue @drmichelle-larue, Jason Bittel @bittelmethis, Kelly Brenner @KellyBrenner, Tianna Burke @Tingo_89, Anthony Caravaggi @thonoir, Alex Evans @alexevans91, Yara Haridy @Yara_Haridy, and Danielle Rivet @grizzlygirl87 for MMM combatant reveals in your Twitter games. The scientist-narrators thank the Simpsons, the Neverending Story, Lord of the Rings, Star Wars Episodes 4–8 in general and Kylo Ren specifically, Basil Stag Hare, Lady Amber, and Skipper, Yote Desert Dog of the Southwest, Pika Jo Varner, Mario-Kart, and baseball lingo. The authors thank Rachel Smythe for Lore Olympus because it is awesome. We thank Joanne Manaster for providing the first media coverage of the tournament in inaugural year 2013, SkunkBear for doing an interview about MMM for NPR Morning Edition in 2015, and John Mead for sharing info about educator resources on NatGeo Education blog in 2018. To all the amplifiers, journalists, educators, students AND ALL PLAYERS, thank you for making March Mammal Madness the transformative community we all deserve. We lastly, and most importantly, acknowledge that the majority of our MMM team members work within settler/colonialist institutions and live and research on stolen land. Science and education anchored to Eurocentrism is impoverished by the exclusion and marginalization of Indigenous people and traditional knowledge, among other deficits of justice, equity, diversity, and inclusion. Eurocentric conservation efforts are similarly inadequate, and we direct readers, and ourselves, to support Indigenous-led environmental organizations

including the Indigenous Environmental Network and Honor the Earth. Full responsibility for the content of this acknowledgement rests with the author team, but we thank Katherine Crocker, Savannah Martin and others for their valuable expertise, time, and insights into the importance, context, and limitations of land acknowledgements.

**Katie Hinde** is in the School of Human Evolution and Social Change, the Center for Evolution and Medicine, and the School of Sustainability, Arizona State University, Tempe, United States; the Brain, Mind, & Behavior Unit, California National Primate Research Center, Davis, United States; and was in the Department of Human Evolutionary Biology, Harvard University, Cambridge, United States 2011-2015
katiehinde@gmail.com
https://orcid.org/0000-0002-0528-866X

**Carlos Eduardo G Amorim** is in the Department of Biology, California State University Northridge, Northridge, United States; and the Department of Computational Biology, University of Lausanne, Lausanne, Switzerland
https://orcid.org/0000-0002-8827-238X

**Alyson F Brokaw** is in the Interdisciplinary Program in Ecology and Evolutionary Biology, Department of Biology, Texas A&M University, College Station, United States
https://orcid.org/0000-0003-3012-1623

**Nicole Burt** is in the Department of Human Health and Evolutionary Medicine, Cleveland Museum of Natural History, Cleveland, United States
https://orcid.org/0000-0003-4453-4808

**Mary C Casillas** is an illustrator based in Dallas, United States. Her work can be seen at: https://marycasillas.wix.com/paintings
https://orcid.org/0000-0002-5421-4341

**Albert Chen** is in the Milner Centre for Evolution, University of Bath, Bath, United Kingdom and the Department of Earth Sciences, University of Cambridge, Cambridge, United Kingdom
https://orcid.org/0000-0002-2671-9190

**Tara Chestnut** is in the National Park Service, Mount Rainier National Park, United States; and the Department of Fisheries and Wildlife, Oregon State University, Corvallis, United States
https://orcid.org/0000-0003-1009-1797

**Patrice K Connors** is in the Department of Biological Sciences, Colorado Mesa University, Grand Junction, United States
https://orcid.org/0000-0002-3816-1585

**Mauna Dasari** is in the Department of Biological Sciences, University of Notre Dame, United States
https://orcid.org/0000-0002-1956-2500

**Connor Fox Ditelberg** is in the Department of Visual & Media Arts, Emerson College, Boston, United States

**Jeanne Dietrick** is at BE Creative LLC, Taylor Mill, United States

**Josh Drew** is in the Department of Ecology, Evolution and Environmental Biology, Columbia University, New York, United States; the Department of Vertebrate Zoology, American Museum of Natural History, New York, United States; and the Department of Environmental and Forest Biology, SUNY College of Environmental Science and Forestry, Syracuse, United States
https://orcid.org/0000-0001-9072-0885

**Lara Durgavich** is in the Department of Human Evolutionary Biology, Harvard University, Cambridge, United States; the Department of Anthropology, Boston University, Boston, United States; and the Department of Anthropology, Tufts University, Medford, United States
https://orcid.org/0000-0003-3024-2900

**Brian Easterling** is at BE Creative LLC, Taylor Mill, United States

**Charon Henning** is an illustrator based in New England, United States. Her work can be seen at: http://www.charonhenning.com

**Anne Hilborn** is in the Department of Evolution, Ecology, and Organismal Biology, University of California Riverside, Riverside, United States
https://orcid.org/0000-0001-9504-1080

**Elinor K Karlsson** is in the Program in Bioinformatics and Integrative Biology, University of Massachusetts Medical School, Worcester, United States; and the Broad Institute of MIT and Harvard, Cambridge, United States

**Marc Kissel** is in the Department of Anthropology, Appalachian State University, Boone, United States; and the Department of Anthropology, University of Notre Dame, Notre Dame, United States
https://orcid.org/0000-0002-4004-1996

**Jennifer Kobylecky** is at the Aldo Leopold Foundation, Baraboo, United States
https://orcid.org/0000-0003-4328-1618

**Jason Krell** is in the Center for Science and Imagination, Arizona State University, Tempe, United States

**Danielle N Lee** is in the Department of Biological Sciences, Southern Illinois University Edwardsville, Edwardsville, United States
https://orcid.org/0000-0002-0488-7214

**Kate M Lesciotto** is in the Department of Clinical Anatomy, College of Osteopathic Medicine, Sam Houston State University, Huntsville, United States; and the Department of Anthropology, Pennsylvania State University, State College, United States
https://orcid.org/0000-0001-9537-5750

**Kristi L Lewton** is in the Department of Integrative Anatomical Sciences, Keck School of Medicine, University of Southern California, Los Angeles, United States; the Department of Mammalogy, Natural History Museum of Los Angeles County, Los Angeles, United States; the Department of Anatomy &

Neurobiology, Boston University School of Medicine, Boston, United States; and the Department of Human Evolutionary Biology, Harvard University, Cambridge, United States
https://orcid.org/0000-0003-0674-2454

**Jessica E Light** is in the Department of Ecology and Conservation Biology, the Biodiversity Research and Teaching Collections, and the Interdisciplinary Program in Ecology and Evolution, Texas A&M University, College Station, United States
https://orcid.org/0000-0001-6462-3045

**Jessica Martin** is in the School of Human Evolution and Social Change, Arizona State University, Tempe, United States

**Asia Murphy** is in the Department of Ecosystem Science and Management, Huck Institutes of the Life Sciences, Pennsylvania State University, State College, United States

**William Nickley** is in the Department of Design, The Ohio State University, Columbus, United States
https://orcid.org/0000-0001-6120-9164

**Alejandra Núñez-de la Mora** is in the Instituto de Investigaciones Psicológicas, Universidad Veracruzana, Xalapa, Mexico
https://orcid.org/0000-0002-1609-0771

**Olivia Pellicer** is an illustrator based in Atlanta, United States. Her work can be seen at: https://opellisms.com
https://orcid.org/0000-0002-0858-3744

**Valeria Pellicer** is an illustrator based in San Francisco, United States. Her work can be seen at: http://www.vpellicerart.com

**Anali Maughan Perry** is in Engagement & Learning Services, ASU Library, Arizona State University, Tempe, United States
https://orcid.org/0000-0001-7173-4827

**Stephanie G Schuttler** is at the North Carolina Museum of Natural Sciences, Raleigh, United States
https://orcid.org/0000-0001-9523-4448

**Anne C Stone** is in the School of Human Evolution and Social Change, the Center for Evolution and Medicine, and the Institute of Human Origins, Arizona State University, Tempe, United States
https://orcid.org/0000-0001-8021-8314

**Brian Tanis** is in the Department of Biology, Oregon State University-Cascades, Cascades, United States
https://orcid.org/0000-0001-9075-4057

**Jesse Weber** is in the Department of Integrative Biology, University of Wisconsin-Madison, Madison, United States
https://orcid.org/0000-0003-4839-6684

**Melissa Wilson** is in the School of Life Sciences and the Center for Evolution and Medicine, Arizona State University, Tempe, United States
https://orcid.org/0000-0002-2614-0285

**Emma Willcocks** is in the Department of Biology, Brown University, Providence, United States
https://orcid.org/0000-0001-7404-3933

**Christopher N Anderson** is in the Department of Biological Sciences, Dominican University, River Forest, United States
https://orcid.org/0000-0001-9641-853X

*Author contributions:* Katie Hinde, Conceptualization, Resources, Data curation, Formal analysis, Supervision, Investigation, Visualization, Methodology, Writing - original draft, Project administration, Writing - review and editing; Carlos Eduardo G Amorim, Alyson F Brokaw, Nicole Burt, Tara Chestnut, Lara Durgavich, Elinor K Karlsson, Jennifer Kobylecky, Jason Krell, Danielle N Lee, Kate M Lesciotto, Jessica Martin, Asia Murphy, Alejandra Núñez-de la Mora, Jesse Weber, Melissa Wilson, Resources, Writing - review and editing; Mary C Casillas, Albert Chen, William Nickley, Olivia Pellicer, Valeria Pellicer, Visualization, Writing - review and editing; Patrice K Connors, Anne Hilborn, Jessica E Light, Stephanie G Schuttler, Resources, Writing - original draft, Writing - review and editing; Mauna Dasari, Resources, Visualization, Writing - original draft, Writing - review and editing; Connor Fox Ditelberg, Emma Willcocks, Data curation, Writing - review and editing; Jeanne Dietrick, Brian Easterling, Resources, Visualization, Writing - review and editing; Josh Drew, Conceptualization, Resources, Methodology, Writing - original draft, Writing - review and editing; Charon Henning, Conceptualization, Resources, Supervision, Visualization, Methodology, Writing - original draft, Project administration, Writing - review and editing; Marc Kissel, Resources, Data curation, Writing - original draft, Writing - review and editing; Kristi L Lewton, Conceptualization, Resources, Methodology, Writing - review and editing; Anali Maughan Perry, Conceptualization, Resources, Data curation, Software, Supervision, Writing - original draft, Writing - review and editing; Anne C Stone, Resources, Supervision, Writing - review and editing; Brian Tanis, Resources, Data curation, Visualization, Writing - review and editing; Christopher N Anderson, Conceptualization, Resources, Supervision, Methodology, Project administration, Writing - review and editing

*Competing interests:* Mary C Casillas, Charon Henning, Olivia Pellicer, Valeria Pellicer: All revenue generated by the sale of tournament artwork through the Society6 shop (https://society6.com/mammalmadness) is equitably divided among the artistic team. Jeanne Dietrick: is an employee of BE Creative LLC. Brian Easterling: is the owner of BE Creative LLC. The other authors declare that no competing interests exist.

*Ethics:* Human subjects: Surveys of educators were conducted after study design and procedures were evaluated for human subjects ethical review under Protocol STUDY00007542 "March Mammal Madness in the Classroom" by PI Katie Hinde approved 1/9/2018

by Arizona State University Institutional Review Board and Modification MOD00009767 approved 2/13/2019.

## Funding

No external funding was received for this work.

### Decision letter and Author response

Decision letter https://doi.org/10.7554/eLife.65066.sa1
Author response https://doi.org/10.7554/eLife.65066.sa2

## Additional files

### Supplementary files

- Supplementary file 1. Play-by-Play Nimravid vs. Tiger Quoll.
- Supplementary file 2. Play-by-Play Springhare vs. Jackrabbit.
- Supplementary file 3. Brief "sports-style" summaries of battle narrations.
- Supplementary file 4. MMM Lesson Plans.
- Supplementary file 5. MMM Worksheets.
- Supplementary file 6. 2018 Educator Survey Instrument.
- Supplementary file 7. 2019 Educator Survey Instrument.
- Transparent reporting form

### Data availability

Data Availability: Source data are publicly available in the ASU Research Data Repository at dataverse.asu.edu/dataverse/marchmammalmadness (Hinde 2021a&b) and linked with the March Mammal Madness Open Resources Collection (Perry and Hinde 2020).

The following datasets were generated:

| Author(s) | Year | Dataset URL | Database and Identifier |
|-----------|------|-------------|-------------------------|
| Hinde K | 2021 | https://doi.org/10.48349/ASU/XTOIAD | ASU Research Data Repository, 10.48349/ASU/XTOIAD |
| Hinde K | 2021 | https://doi.org/10.48349/ASU/KKXMSF | ASU Research Data Repository, 10.48349/ASU/KKXMSF |

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
