## [Decision Letter]

Thank you for submitting your article "March Mammal Madness and the Power of Narrative in Science Outreach" to *eLife* for consideration as a Feature Article. Your article has been reviewed by three peer reviewers, including George Perry as the Senior and Reviewing Editor and Reviewer #3. The following individuals involved in review of your submission have agreed to reveal their identity: Penny Bishop (Reviewer #1); Michelle Bezanson (Reviewer #2).

The reviewers and editors have discussed the reviews and we have drafted this decision letter to help you prepare a revised submission.

Summary:

This is an outstanding contribution that describes the motivation, mechanics, adoption, and impact of a leading science education activity, March Mammal Madness. The manuscript is written in a highly-engaging style while covering a lot of ground. That is, readers will find continuous points of interest whether they are fellow science communicators, familiar March Mammal Madness participants, educators, or those who will be introduced to this activity via this article, all without losing the sense of the artistry, excitement, and inspiration that embody March Mammal Madness (and this manuscript). Well done.

Essential revisions:

I am including the full reviews from all three reviewers. In consultation we agreed that all of the (relatively small number of) points raised within the reviews should be considered as essential revisions. Key points include appropriately distinguishing between educator and student perspectives and more directly addressing the operational definition of impact used in this report (and probably, also discussing this on a more expansive basis), and explicitly highlighting the important role of natural history and observational science in the March Mammal Madness ecosystem. See below for additional important points to address in your revision.

Reviewer #1:

This paper is a well-conceptualized documentation of a novel approach to science education, namely a tournament, housed on social media, of research-based, hypothetical encounters between mammal "combatants". It would be of broad interest to scientists interested in new avenues for outreach, as well as to middle and high school educators. The manuscript's strengths include interesting and unusual subject matter, an engaging and accessible tone, and solid substantiation of claims.

Thank you for the opportunity to review your work. I learned a great deal about MMM, an initiative with which I was unfamiliar previously. You provided a well-organized and detailed accounting of a truly novel approach to science education and outreach. I particularly appreciated the engaging, and even at times humorous, tone, which seems all too rare in scholarly writing. As an educator, I also appreciated your focus on how narrative arcs can facilitate learning, and how gamification, storytelling, and social media can be leveraged towards those ends.

By documenting the design, you offered appropriate and sufficient context for readers who are unfamiliar with the tournament to understand its aims and approaches. You also effectively established the reach of the tournament, elaborating the nature of the social media community, and the increase in user engagement as documented through blogpost views and Twitter hashtag use, among other measures. The development and use of educational resources associated with the tournament were similarly compelling ways to consider impact.

I have a few minor suggestions and questions for your consideration that you may find useful.

1) How you are defining impact for the purposes of this paper is somewhat unclear. Based on some of the data you share, it appears connected to teachers' perceptions of student engagement, whereas educators and the public might assume student learning to be a goal. You make reference to broader impacts related to science communication. I would encourage you to be more explicit about this.

2) The reliance on teacher perceptions of student experience, rather than students' voicing their own experience, is a fairly common shortcoming in educational research. If you have student-level data, consider adding it. If you don't, consider naming this issue or limitation. Relatedly, the "Emergent Community" section provides lively examples of the type of community that MMM has fostered. I am left to wonder how much of the "fandom" described is comprised of the adult community and what role adolescents in fact play in it, how they perceive it, etc. (I do recognize this is not a central aim of the paper, however. ) Further, the proliferation of laudatory comments from educators regarding their students' experience are impressive. They might be even more compelling, and perceived as more trustworthy, if they were offset by a brief consideration of concomitant challenges or the provision of any negative data.

3) You use first person plural as the authors and it is then that I recognized that the authors are the same as the MMM team. I suggest identifying this positionality earlier on.

4) You state reference potential "chaotic derailment in a classroom of adolescents," which in my opinion underscores society's deficit perspective of adolescents.

Overall, I found the manuscript to be very well written and of value to a broad educator audience. (As an aside, I'll be recommending MMM to my own professional networks as a result of the opportunity to learn about it!).

Reviewer #2:

In this paper, the authors eloquently and thoroughly describe the science outreach tournament Mammals March Madness (MMM). Full disclosure: I have been a fan of this tournament since the very beginning and feel it might be one of the most successful animal behavior educational models today. The paper is well-written, detailed, and provides examples that are certain to provide opportunities for role modeling for many generations of STEM learners and educators. The manuscript will have a large impact due to excellent prose and the fact that MMM is interdisciplinary not just within the natural sciences, but with science art as well. I tried, but could not find any weaknesses in the paper.

It is clear that much work and review has already been accomplished in this draft of the manuscript. I think the length, figures, and Materials and methods are clear and appropriate. I truly think this is ready to go. If I am pushed to make one suggestion it is the following.

I think it would be useful to highlight the importance of natural history in a similar fashion (shorter is fine) as the narrative. MMM brings us back to the importance natural history and observational data. We do not get told that observational science, narratives, natural history is a possibility in our science classes. We get forced into ideas about experimental science for science fair. At the big intel science fair, from 2014-2019, there were 379 entries in the Animal Science category. Seven of these were natural history or observational studies that one could do outside and free of charge. It is worth noting that many of the observational studies focused on conservation and won prizes. So, that is about 1 of 63 projects.

There has been an abysmally disappointing and sad decline in natural history (NH) research and this may parallel a decline in public participation in nature (Tewksbury et al., 2014). In 1950s: all schools had NH, now majority do not and NH emphasis is declining in text books. Organismal classes are missing from college curricula. These are classes that focus on whole organisms: ornithology, mammology, herpetology. I think this manuscript has an opportunity to highlight this in a brief paragraph and discuss how MMM can contribute to changing instructors and learners ideas about organismal biology and natural history work. If this is to be integrated, I also recommend looking at Harry Greene's work to add to the citation list.

Reviewer #3:

This manuscript from Hinde et al. is a tour de force report of the development, mechanics, and impact of one of the most amazing science communication initiatives of our time, March Mammal Madness (MMM). This stands to serve as (1) a companion guide for educators not yet incorporating MMM activities into their lessons, (2) an enrichment for those educators who already have, (3) an incredibly enjoyable read for all long-term fans of the event and those interested in science communication, and (4) an important contribution to science communication literature. The statistics reporting on the level of engagement are astounding, and it is special to see these data side by side with explanations of how MMM operates, the science communication goals as told from the perspectives of the authors, and with the voices of educators who have incorporated MMM into their classrooms. And of course with the amazing art! Both the visual (e.g. Figures 5 and 8) and the narrative (as emphasized effectively throughout) art.

I have very little substantive critique to offer, as in my view this manuscript already is a masterpiece.

1) I valued the discussion beginning with "Since 2014, the organizers have intentionally designed divisions to integrate more complex themes of environments, extinction-risk, adaptations…" because this offered some insight into the thoughtful design into the tournament as connected to its base (and understandably continually developing) science communication aims. Prior to this point in the manuscript, the sense of purposeful decision-making by the authors was perhaps a bit too between-the-lines, too passive. This could be addressed very concisely by prefacing the later-to-come insight, at an earlier point.

2) The perspectives on impact (both from survey data and as represented in the ethnographic data and illustrative quotes) are from educators rather than perhaps the ultimate target audience, the child and young adult learners. The educator-framed results and insights are still incredibly valuable!, but this limitation and any considerations for potential future impact and learning effectiveness assessments, etc., could be discussed more explicitly.

3) In the interest of ensuring full transparency, I believe that a COI should be reported for Charon Henning (and any other contributing artists who are co-authors) re: the revenue from the sales of the MMM art, which is discussed in the manuscript.

---

## [Author Response]

Reviewer #1:Thank you for the opportunity to review your work. I learned a great deal about MMM, an initiative with which I was unfamiliar previously. You provided a well-organized and detailed accounting of a truly novel approach to science education and outreach. I particularly appreciated the engaging, and even at times humorous, tone, which seems all too rare in scholarly writing. As an educator, I also appreciated your focus on how narrative arcs can facilitate learning, and how gamification, storytelling, and social media can be leveraged towards those ends.By documenting the design, you offered appropriate and sufficient context for readers who are unfamiliar with the tournament to understand its aims and approaches. You also effectively established the reach of the tournament, elaborating the nature of the social media community, and the increase in user engagement as documented through blogpost views and Twitter hashtag use, among other measures. The development and use of educational resources associated with the tournament were similarly compelling ways to consider impact.

Thank you so very much. We wanted our manuscript to reflect our tournament by weaving scholarly content with elements of humor. We agree that this is too rare in scholarly writing and speculate that positioning scholars and scholarship as largely humorless is an unforced error that widens the chasm among scientists and publics.

I have a few minor suggestions and questions for your consideration that you may find useful.1) How you are defining impact for the purposes of this paper is somewhat unclear. Based on some of the data you share, it appears connected to teachers' perceptions of student engagement, whereas educators and the public might assume student learning to be a goal. You make reference to broader impacts related to science communication. I would encourage you to be more explicit about this.

Your point is excellent. To better situate impact in our manuscript, we have expanded the Introduction, Results, and Discussion sections (and added ~20 citations). We now more explicitly address how impact has been assessed for SciComm programs in the Introduction, situated educator competence in assessing student achievement and engagement when discussing surveys limitations in the Educational Resources section, and explained how student engagement as a multifactorial construct contributes, in part, to learning outcomes in the broader section on Narrative Facilitates Learning and Human Adaptations at Play.

2) The reliance on teacher perceptions of student experience, rather than students' voicing their own experience, is a fairly common shortcoming in educational research. If you have student-level data, consider adding it. If you don't, consider naming this issue or limitation. Relatedly, the "Emergent Community" section provides lively examples of the type of community that MMM has fostered. I am left to wonder how much of the "fandom" described is comprised of the adult community and what role adolescents in fact play in it, how they perceive it, etc. (I do recognize this is not a central aim of the paper, however. ) Further, the proliferation of laudatory comments from educators regarding their students' experience are impressive. They might be even more compelling, and perceived as more trustworthy, if they were offset by a brief consideration of concomitant challenges or the provision of any negative data.

This is a very important set of considerations, thank you for presenting them.

While we have an extensive collection of tweets by engaged students that are available often publicly, particular favorites are kids tweeting that they’ve asked their parents’ permission to stay up past their bedtime to “watch” the twitter battles, student letters requesting particular animal combatants, and photos of learners with brackets conducting research, the identifiable elements and underage status precludes us from treating these as data. Instead we have added text to highlight this limitation and how to address it with future research in the discussion of the limitations of the surveys (more about this below).

In response to the recommendation for a balanced presentation of “laudatory comments” and “negative data”, we coded the final free-write comments from educators as positive, negative, constructive, combination, or other. In the final “comments” about the experiences of using March Mammal Madness with their learners, ~90% of responses included positive content, fewer than 4% of comments included negative content. In addition to reporting this in the text (and updating the Materials and methods), readers can access all comments via the source data in our dataverse repository.

Further, we have added a paragraph that unpacks the primary limitations of our survey design including selection bias, indirect access and aggregated representation of student experiences, and unknown learning outcomes. In this paragraph we further identified future research directions for addressing these limitations.

3) You use first person plural as the authors and it is then that I recognized that the authors are the same as the MMM team. I suggest identifying this positionality earlier on.

To better situate ourselves as the creative team behind the tournament, the Abstract and Introduction now includes more intentional and consistent use of possessive pronouns.

4) You state reference potential "chaotic derailment in a classroom of adolescents," which in my opinion underscores society's deficit perspective of adolescents.

Thank you for bringing this to our attention. This statement was made as an allusion to tweets from multiple educators who have communicated that some of our combatants or tournament events created notable disruption in their classrooms (students having laughing fits) that has shaped our planning process. Our aim was to advise other scientists to consider the developmental stage(s)/fascinations of audiences when planning public communication and to consider how materials may play out for educators trying to manage their classrooms. We have deleted the phrase “chaotic derailment” and revised this text to state “minimize counter-productive digressions in classrooms of adolescents in contexts of various cultural sensibilities.”

Overall, I found the manuscript to be very well written and of value to a broad educator audience. (As an aside, I'll be recommending MMM to my own professional networks as a result of the opportunity to learn about it!).

Thank you. We are really excited for the 2021 Tournament.

Reviewer #2:In this paper, the authors eloquently and thoroughly describe the science outreach tournament Mammals March Madness (MMM). Full disclosure: I have been a fan of this tournament since the very beginning and feel it might be one of the most successful animal behavior educational models today. The paper is well-written, detailed, and provides examples that are certain to provide opportunities for role modeling for many generations of STEM learners and educators. The manuscript will have a large impact due to excellent prose and the fact that MMM is interdisciplinary not just within the natural sciences, but with science art as well. I tried, but could not find any weaknesses in the paper.It is clear that much work and review has already been accomplished in this draft of the manuscript. I think the length, figures, and Materials and methods are clear and appropriate. I truly think this is ready to go. If I am pushed to make one suggestion it is the following.I think it would be useful to highlight the importance of natural history in a similar fashion (shorter is fine) as the narrative. MMM brings us back to the importance natural history and observational data. We do not get told that observational science, narratives, natural history is a possibility in our science classes. We get forced into ideas about experimental science for science fair. At the big intel science fair, from 2014-2019, there were 379 entries in the Animal Science category. Seven of these were natural history or observational studies that one could do outside and free of charge. It is worth noting that many of the observational studies focused on conservation and won prizes. So, that is about 1 of 63 projects.There has been an abysmally disappointing and sad decline in natural history (NH) research and this may parallel a decline in public participation in nature (Tewksbury et al., 2014). In 1950s: all schools had NH, now majority do not and NH emphasis is declining in text books. Organismal classes are missing from college curricula. These are classes that focus on whole organisms: ornithology, mammology, herpetology. I think this manuscript has an opportunity to highlight this in a brief paragraph and discuss how MMM can contribute to changing instructors and learners ideas about organismal biology and natural history work. If this is to be integrated, I also recommend looking at Harry Greene's work to add to the citation list.

Thank you for this opportunity to more intentionally situate natural history within our manuscript and guidance for key papers/authors as we initiated our literature search. We’ve added 7 citations addressing the importance, decline, and revitalization of natural history within the sciences. As we approached the revision, we found that these points/citations worked particularly well embedded within existing paragraphs- specifically in sections on combatant species and scholarly literature featured in MMM, why narrative enhances learning, reintegrating art into science curricula, Indigenous knowledge, and lasting impacts. We hope that this integration achieves the goal of highlighting natural history and how MMM fills unfortunate gaps in organism-focused curricula.

Reviewer #3:This manuscript from Hinde et al. is a tour de force report of the development, mechanics, and impact of one of the most amazing science communication initiatives of our time, March Mammal Madness (MMM). This stands to serve as (1) a companion guide for educators not yet incorporating MMM activities into their lessons, (2) an enrichment for those educators who already have, (3) an incredibly enjoyable read for all long-term fans of the event and those interested in science communication, and (4) an important contribution to science communication literature. The statistics reporting on the level of engagement are astounding, and it is special to see these data side by side with explanations of how MMM operates, the science communication goals as told from the perspectives of the authors, and with the voices of educators who have incorporated MMM into their classrooms. And of course with the amazing art! Both the visual (e.g. Figures 5 and 8) and the narrative (as emphasized effectively throughout) art.I have very little substantive critique to offer, as in my view this manuscript already is a masterpiece.1) I valued the discussion beginning with "Since 2014, the organizers have intentionally designed divisions to integrate more complex themes of environments, extinction-risk, adaptations…" because this offered some insight into the thoughtful design into the tournament as connected to its base (and understandably continually developing) science communication aims. Prior to this point in the manuscript, the sense of purposeful decision-making by the authors was perhaps a bit too between-the-lines, too passive. This could be addressed very concisely by prefacing the later-to-come insight, at an earlier point.

Thank you for bringing to attention to the vague intentionality and passive non-attribution that characterizes the prose, especially in the early sections of the manuscript. As background, the tournament was launched as a lark in 2013, with the game structure of battles spontaneously created by KH on a Friday afternoon in the lead-up to departmental happy hour. With each subsequent year, as folks with diverse skills sets became enamored with the tournament, they often spontaneously volunteered their contributions. As Editor-in-Chief, KH would coordinate balance and structure across these contributors and elements, but in so many ways March Mammal Madness was a collective, emergent phenomenon. As we wrote this manuscript and revisited literatures (social learning, narrative structure cognition, content biases) or discovered literatures (sports fan engagement, shared experiences at festivals) it became evident to us that we had wonderfully, but somewhat unwittingly, tapped into dimensions of human cognition and psychology.

We have revised the manuscript to address this issue in several ways. The inclusion of more possessive pronouns in early sections in response to reviewer #1 required some more active and intentional phrasing. Similarly, in response to reviewer #2, as we integrated the importance of natural history, we were able to more explicitly discuss intentional planning in species selections. We revised in several other sections to reflect intentional planning and applied design. Lastly we made some adjustments in the timeline section to more explicitly situate serendipity and the emergent “alchemy” of March Mammal Madness.

2) The perspectives on impact (both from survey data and as represented in the ethnographic data and illustrative quotes) are from educators rather than perhaps the ultimate target audience, the child and young adult learners. The educator-framed results and insights are still incredibly valuable!, but this limitation and any considerations for potential future impact and learning effectiveness assessments, etc., could be discussed more explicitly.

This recommendation replicates several points from reviewer #1 and we appreciate the importance of addressing these issues. Thank you for advocating that we better consider learners even if we do not yet have direct, systematic data at this time. Briefly to flashback to our response to reviewer #1, we have revised the manuscript to better situate March Mammal Madness and our educator surveys within constructs of impact and user engagement, while discussing the limitations and future approaches to more directly assess learners’ perceptions and outcomes. We have revised the text to note “illustrative” quotes.

3) In the interest of ensuring full transparency, I believe that a COI should be reported for Charon Henning (and any other contributing artists who are co-authors) re: the revenue from the sales of the MMM art, which is discussed in the manuscript.

Corrected at revision resubmission.